# InvarGC: Invariant Granger Causality for Heterogeneous Interventional Time Series under Latent Confounding

## Abstract

Granger causality is widely used for causal structure discovery in complex systems from multivariate time series data. Traditional Granger causality tests based on linear models often fail to detect even mild non-linear causal relationships. Therefore, numerous recent studies have investigated non-linear Granger causality methods, achieving improved performance. However, these methods often rely on two key assumptions: causal sufficiency and known interventional targets. Causal sufficiency assumes the absence of latent confounders, yet their presence can introduce spurious correlations. Moreover, real-world time series data usually come from heterogeneous environments, without prior knowledge of interventions. Therefore, in practice, it is difficult to distinguish intervened environments from non-intervened ones, and even harder to identify which variables or timesteps are affected. To address these challenges, we propose Invariant Granger Causality (InvarGC), which leverages cross-environment heterogeneity to mitigate the effects of latent confounding and to distinguish intervened from non-intervened environments with edge-level granularity, thereby recovering invariant causal relations. In addition, we establish the identifiability under these conditions. Extensive experiments on both synthetic and real-world datasets demonstrate the competitive performance of our approach compared to state-of-the-art methods.

## 1 Introduction

Granger causality has been widely used to uncover causal relationships in time series data in various real-world applications, including finance, healthcare, and retail pricing. Traditional Granger causality tests based on linear models often struggle to capture even subtle non-linear causal dependencies (Granger, 1969). With the emergence and advancement of neural networks (LeCun et al., 2015), significant research efforts have been dedicated to improving Granger causality methods to account for non-linearities (Khanna & Tan, 2019; Marcinkevičs & Vogt, 2021; Tank et al., 2021; Cheng et al., 2023; Zhou et al., 2024; Cheng et al., 2024; Han et al., 2025).

Although non-linear Granger causality provides a more flexible framework for capturing complex causal relationships, most existing methods for learning Granger causality still struggle due to their reliance on the assumption of causal sufficiency (Pearl, 2009; Perry et al., 2022; Wang & Drton, 2023; Reddy & N Balasubramanian, 2024). When latent confounding presents and causal sufficiency does not hold, these methods fail to accurately identify Granger causality, as they do not account for hidden variables that may influence multiple observational variables (Geiger et al., 2015; Malinsky & Spirtes, 2018). This limitation underscores the need for more advanced approaches that can effectively handle latent confounding and infer Granger causality in their presence.

An even greater challenge is inferring Granger causality under both latent confounding and unknown interventions, where only time series data collected from multiple environments are available, without labels indicating whether an environment is intervened or non-intervened, nor which variables and time steps are targeted by interventions. Existing methods either assume that the time series are stationary and thus overlook interventions (Huang et al., 2020), rely on labels indicating which environments are non-intervened (Han et al., 2025), or assume that the timing and targets of interventions are known (Gao et al., 2022; Liu & Kuang, 2023). However, these assumptions are often

unrealistic in real-world scenarios (Squires & Uhler, 2023; Mameche et al., 2024). In practice, it is difficult to distinguish intervened environments from non-intervened ones, and to identify which variables are targeted and when (Brouillard et al., 2020; Perry et al., 2022). Latent confounders further complicate intervention identification, as they may induce spurious intervention-like effects or hide actual intervention signals. Addressing these challenges requires methodologies that, without any prior knowledge of interventions, can distinguish intervened from non-intervened environments while mitigating latent confounding to enable reliable Granger causality inference.

In this paper, we first reformulate the framework of Granger causality to explicitly address the challenges posed by latent confounding and unknown interventions. To this end, we propose Invariant Granger Causality (InvarGC), a novel framework that operates on heterogeneous interventional time series data under latent confounding, which leverages environmental heterogeneity to both distinguish intervened environments from non-intervened ones and mitigate latent confounding, thereby recovering invariant causal relations across environments. The main contributions of this work can be summarized as follows:

- **Problem Formulation.** We reformulate the framework of Granger causality to explicitly address the challenges posed by latent confounding and unknown interventions, which are prevalent in real-world time series but largely overlooked in existing methods.

- **Methodology.** We propose InvarGC, which leverages environmental heterogeneity to simultaneously mitigate latent confounding, distinguish intervened from non-intervened environments, identify edge-level interventions, and recover invariant causal structures.

- **Theoretical Guarantee.** We establish identifiability results for InvarGC, showing that the Granger causal graph, latent confounder subspace, and node-/edge-level interventions can be consistently recovered under appropriate assumptions.

- **Empirical Validation.** Extensive experiments on synthetic and real-world datasets demonstrate that InvarGC outperforms robust baselines in accuracy and interpretability, even under latent confounding and unknown interventions.

## 2 RELATED WORK

**Granger Causality-based Methods.** Linear Granger causality-based methods are mostly built upon regularized vector autoregressive (VAR) models (Granger, 1969). Arnold et al. (2007) first introduced Granger causality inference using LASSO (Tibshirani, 1996), while Tank et al. (2021) extended this approach to non-linear setting through a sparse-input MLP and LSTM. Khanna & Tan (2019) leveraged the efficiency of the economical statistical recurrent unit (eSRU) architecture, incorporating Group-LASSO (Yuan & Lin, 2006) regularization at the input layer. Marcinkevičs & Vogt (2020) proposed a generalized vector autoregressive (GVAR) model that improves neural network interpretability with sign detection. Cheng et al. (2023; 2024) addresses a challenging setting and developed approaches to infer Granger causality from irregular time series data. Zhou et al. (2024) introduced a Jacobian regularizer-based framework that constructs a single efficient model to predict all target variables simultaneously. Han et al. (2025) proposed an encoder-decoder architecture that effectively leverages anomalous time series data to infer Granger causality. However, these existing methods assume causal sufficiency and thus overlook the effects of latent confounding.

**Methods for Latent Confounding and Interventions.** When causal sufficiency does not hold, general algorithms such as the Fast Causal Inference (FCI) family (Spirtes et al., 2000; Zhang, 2008; Colombo et al., 2012; Ogarrio et al., 2016) and optimization-based approaches (Chandrasekaran et al., 2010) can detect confounding in limited contexts. In the time series domain, tsFCI (Entner & Hoyer, 2010) adapts FCI to sliding time windows, SVAR-FCI (Malinsky & Spirtes, 2018) incorporates stationarity assumptions to refine edge orientation, and LPCMCI (Gerhardus & Runge, 2020) extends PCMCI (Runge et al., 2019) to account for hidden confounders, providing clearer causal interpretations. In the context of interventions, prior work in static settings has investigated combining observational and interventional data (Yang et al., 2018) and analyzing perfect or imperfect interventions with known or unknown targets (Brouillard et al., 2020). More recent studies further extend these efforts to non-stationary time series, including latent intervened domain recovery (Liu & Kuang, 2023) and joint learning from observational and interventional time series (Gao et al., 2022), yet none of existing methods jointly address latent confounders and unknown interventions.

## 3 PRELIMINARIES

### 3.1 NON-LINEAR GRANGER CAUSALITY

Granger causality was originally defined for linear relationships within vector autoregressive processes (VAR) to model causal relationships in multivariate time series data. More recently, methods to capture non-linear Granger causality have been developed using neural network. Consider a stationary multivariate time series $X = \{X_1, \ldots, X_T\}$ with $T$ timesteps, where each $X_t \in \mathbb{R}^d$. Assume that causal relationships between variables are given by the following structural model:

$$X_{t+1}^i = f_i(X_{1:t}^1, \ldots, X_{1:t}^d) + \epsilon_{t+1}^i \quad \text{for } 1 \le i \le d, \tag{1}$$

where $f_i$ is a function that specifies how the past values are mapped to time series $i$. Time series $j$ is Granger non-causal for time series $i$ if $f_i(X_{1:t}^1, \ldots, X_{1:t}^j, \ldots, X_{1:t}^d) = f_i(X_{1:t}^1, \ldots, \tilde{X}_{1:t}^j, \ldots, X_{1:t}^d)$ for all $X_{1:t}^1, \ldots, X_{1:t}^d$ and all $X_{1:t}^j \ne \tilde{X}_{1:t}^j$.

### 3.2 STRUCTURAL CAUSAL MODEL WITH LATENT CONFOUNDERS IN TIME SERIES

The standard linear structural causal models (SCMs) for a set of observed variables $X$, assuming no latent confounders, can be described as $X = W^\top X + \epsilon$, where $W$ is the weighted adjacency matrix encoding the causal relationships among variables in $X$, and each noise term $\epsilon_i$ is assumed to be independent of the parents $\text{PA}(X_i)$ of variable $X_i$. If we further assume no instantaneous effects in time series, the SCMs can be adapted into a first-order vector autoregressive form, $X_{t+1} = W^\top X_t + \epsilon_{t+1}$, where $W$ represents the weighted adjacency matrix that captures the causal relationships from $X_t$ to $X_{t+1}$. To incorporate latent confounders in time series, we reformulate $X_t$ as $(X_t, Z_t)$, and

$$\begin{pmatrix} X_{t+1} \\ Z_{t+1} \end{pmatrix} = \begin{pmatrix} W_{X_{t+1}X_t}^\top & W_{X_{t+1}Z_t}^\top \\ 0 & W_{Z_{t+1}Z_t}^\top \end{pmatrix} \begin{pmatrix} X_t \\ Z_t \end{pmatrix} + \begin{pmatrix} \epsilon_{t+1}^X \\ \epsilon_{t+1}^Z \end{pmatrix}, \tag{2}$$

where we assume $\text{PA}(Z_{t+1}) = Z_t$ and $W_{Z_{t+1}X_t} = 0$. This yields the component-wise equations,

$$X_{t+1} = W_{X_{t+1}X_t}^\top X_t + W_{X_{t+1}Z_t}^\top Z_t + \epsilon_{t+1}^X, \tag{3}$$

$$Z_{t+1} = W_{Z_{t+1}Z_t}^\top Z_t + \epsilon_{t+1}^Z. \tag{4}$$

Here, $W_{X_{t+1}X_t}$ encodes the Granger causality in multivariate time series data $X$, while $W_{X_{t+1}Z_t}$ captures the causal relationships from $Z_t$ to $X_{t+1}$ and $W_{Z_{t+1}Z_t}$ is a diagonal matrix. To flexible this framework, we generalize Eqs.(3–4) to accommodate non-linear settings as follows,

$$X_{t+1} = f_X(W_{X_{t+1}X_t}^\top X_t + W_{X_{t+1}Z_t}^\top Z_t) + \epsilon_{t+1}^X, \tag{5}$$

$$Z_{t+1} = f_Z(W_{Z_{t+1}Z_t}^\top Z_t) + \epsilon_{t+1}^Z. \tag{6}$$

where $f_X$ and $f_Z$ are functions that can be selected from either linear or non-linear classes.

### 3.3 TYPES OF INTERVENTIONS IN TIME SERIES

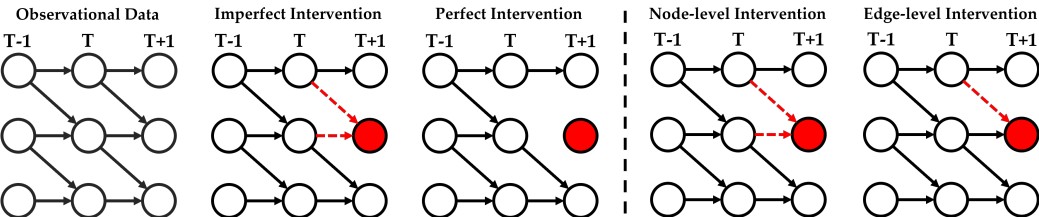

Figure 1: (Left): Imperfect intervention alter all causal relations from the parent nodes to the target node, whereas perfect intervention disconnect the target node from its parents; (Right): Node-level intervention apply to all edges, whereas edge-level intervention generalize to any subset.

In time series, an intervention on a variable $X_t^i$ is defined by replacing its conditional distribution $p(X_t^i|\text{PA}(X_t^i))$ with a modified distribution $\tilde{p}(X_t^i|\text{PA}(X_t^i))$, where $\text{PA}(X_t^i)$ denotes its set of parents.

Broadly, the types of interventions can be classified as imperfect (soft or parametric) interventions, with perfect (hard or structural) interventions being a special case in which all parental influence are entirely removed, i.e., $p(X_t^i|\text{PA}(X_t^i)) = p(X_t^i)$. Interventions can also be categorized by granularity, node-level interventions are applied to a target node and alter or remove all edges connecting it to its parents, and they represent a special case of edge-level interventions, which may affect either all or only a subset of parent–child relations. Both types of categorization are illustrated in Figure 1.

# 4 INVARIANT GRANGER CAUSALITY

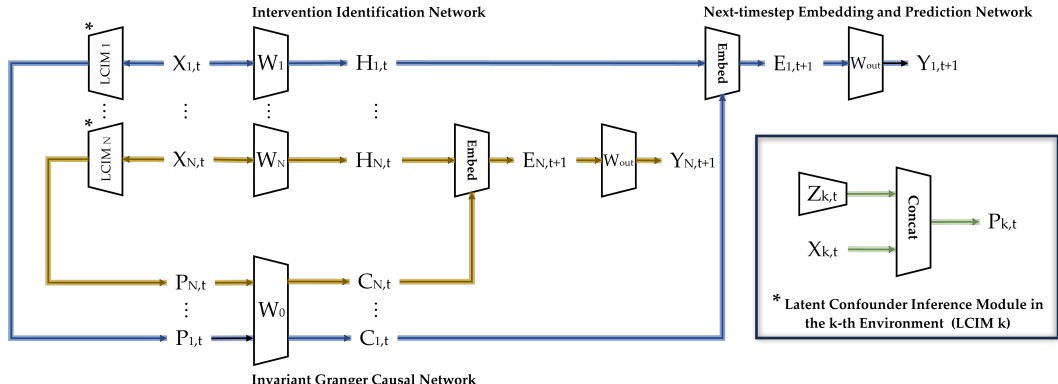

Figure 2: InvarGC is composed of latent confounder inference modules, intervention identification network, invariant Granger causal network, and next-timestep embedding and prediction network.

In this section, we first introduce the InvarGC framework. In general, each environment is equipped with a latent confounder inference module (LCIM) together with an intervention identification network, while all environments share a common invariant Granger causal network. By leveraging cross-environment variation to mitigate the effects of latent confounders and distinguish interventions from purely observational settings, the framework learns an invariant Granger causal structure. We further present the corresponding optimization strategy and finally establish the identifiability of the proposed framework.

## 4.1 MODEL ARCHITECTURE

Given heterogeneous multivariate time series data $\mathbf{X} \in \mathbb{R}^{N \times d \times T}$ collected from $N$ environments, we denote $\mathbf{X} = \{\mathbf{X}_1, \mathbf{X}_2, \ldots, \mathbf{X}_N\}$, where $\mathbf{X}_k \in \mathbb{R}^{d \times T}$ represents the multivariate time series from environment $k$, consisting of $d$ observed variables over $T$ timesteps.

**Latent Confounder Inference Module.** For each environment $k$ and timestep $t$, with observed variables $\mathbf{X}_{k,t} \in \mathbb{R}^d$, we initialize a learnable vector $\mathbf{Z}_{k,t} \in \mathbb{R}^p$, where $p$ denotes the number of latent confounders. The resulting input vector is constructed by concatenating the observed variables with the latent confounders:

$$\mathbf{P}_{k,t} = \text{LCIM}_k(\mathbf{X}_{k,t}, \mathbf{Z}_{k,t}) \in \mathbb{R}^{d+p}. \tag{7}$$

**Intervention Identification Network.** In practice, interventions are generally applied to observed variables; hence for each input $\mathbf{X}_{k,t}$, the intervention identification network applies a linear projection to map it into an intervention representation:

$$\mathbf{H}_{k,t} = \mathbf{W}_k \mathbf{X}_{k,t} \in \mathbb{R}^h. \tag{8}$$

**Invariant Granger Causal Network.** All concatenated inputs $\mathbf{P}_t = \{\mathbf{P}_{1,t}, \mathbf{P}_{2,t}, \ldots, \mathbf{P}_{N,t}\}$ from the LCIMs share a common invariant Granger causal network, which projects each $\mathbf{P}_{k,t}$ into an invariant causal representation:

$$\mathbf{C}_{k,t} = \mathbf{W}_0 \mathbf{P}_{k,t} \in \mathbb{R}^h. \tag{9}$$

**Next-timestep Embedding and Prediction Network.** To generate next-timestep prediction, we combine the intervention representation $\mathbf{H}_{k,t}$ with the invariant causal representation $\mathbf{C}_{k,t}$, and apply an embedding function $\text{Embed}(\cdot)$ to capture potential nonlinear interactions between them:

$$\mathbf{E}_{k,t+1} = \text{Embed}(\mathbf{H}_{k,t}, \mathbf{C}_{k,t}) \in \mathbb{R}^{h'}. \tag{10}$$

The resulting embedding $\mathbf{E}_{k,t+1}$ is linearly projected to generate the next-timestep prediction for the $k$-th environment:

$$\hat{\mathbf{Y}}_{k,t+1} = \mathbf{W}_{\text{out}}\mathbf{E}_{k,t+1} \in \mathbb{R}^d. \tag{11}$$

Figure 2 illustrates the overall architecture and workflow of the proposed framework.

## 4.2 OPTIMIZATION

In the optimization for model training, we first define $P_{k,t} = [X_{k,t}, Z_{k,t}] \in \mathbb{R}^{d+p}$ as the concatenation of observed variables $X_{k,t}$ and latent confounders $Z_{k,t}$. For each variable $i$, we denote by $W_0^i \in \mathbb{R}^{1\times(d+p)}$ the invariant weight vector mapping $P_{k,t}$ to $X_{k,t+1}^i$, thereby capturing time-lagged causal dependencies that remain invariant across environments. In addition, for each environment $k$, we define $W_k^i \in \mathbb{R}^{1\times d}$ as the edge-level intervention weight vector from $X_{k,t}$ to $X_{k,t+1}^i$. With these definitions in place, we propose a unified loss function in Eq.(12) that jointly optimizes over $W$ and $Z$:

$$\min_{W,Z} \sum_{k=1}^{N}\sum_{i=1}^{d}\sum_{t=1}^{T} \left\| X_{k,t+1}^i - (W_0^i P_{k,t} + W_k^i X_{k,t}) \right\|_2^2 + \lambda_z \sum_{k=1}^{N}\sum_{l=1}^{L}\sqrt{\frac{1}{T}\sum_{t=1}^{T}Z_{k,l,t}^2}$$
$$+ (1-\alpha)\sum_{i=1}^{d}\sum_{j=1}^{d+p}\left\|(W_{0,j}^i, W_{1,j}^i, ..., W_{N,j}^i)\right\|_2 + \alpha\sum_{k=1}^{N}\sum_{i=1}^{d}\sum_{j=1}^{d}\left\|W_{k,j}^i\right\|_2, \tag{12}$$

where $L$ is the number of latent confounders to be inferred, $\lambda_z > 0$ is a regularization parameter that penalizes the latent confounder matrix, and $\alpha \in (0,1)$ balances sparsity across groups and within groups. The proposed formulation simultaneously mitigates the effects of latent confounders through the design of $Z$, identifies edge-level interventions $W_k \in \mathbb{R}^{d\times d}$ for each environment, and estimates the invariant Granger causal graph $W_{0,X_{t+1}X_t} \in \mathbb{R}^{d\times d}$, i.e.,the observed-to-observed submatrix of $W_0 \in \mathbb{R}^{d\times(d+p)}$. To extend the formulation to non-linear relationships, we assume the existence of functions $f_i(\cdot)$, $g_{k,i}(\cdot)$, and $h_{k,i}(\cdot)$ such that: $\mathbb{E}[X_{k,t+1}^i|\text{PA}(X_{k,t+1}^i), Z_t] = h_{k,i}\big(f_i(P_{k,t}), g_{k,i}(X_{k,t})\big)$, where $f_i(\cdot)$ denotes the invariant non-linear function that generates the $i$-th variable from its observed causal parents and latent confounders, consistently across all environments, $g_{k,i}(\cdot)$ encodes edge-level interventions acting on variable $i$ in environment $k$, and $h_{k,i}(\cdot)$ acts as an aggregation function, integrating the invariant mechanism $f_i(\cdot)$ with the intervention component $g_{k,i}(\cdot)$. The overall objective is then reformulated as:

$$\min_{W,Z} \sum_{k=1}^{N}\sum_{i=1}^{d}\sum_{t=1}^{T} \left\| X_{k,t+1}^i - h_{k,i}\big(f_i(P_{k,t};W_0^i), g_{k,i}(X_{k,t};W_k^i)\big) \right\|_2^2 + \lambda_z \sum_{k=1}^{N}\sum_{l=1}^{L}\sqrt{\frac{1}{T}\sum_{t=1}^{T}Z_{k,l,t}^2}$$
$$+ (1-\alpha)\sum_{i=1}^{d}\sum_{j=1}^{d+p}\left\|(W_{0,j}^i, W_{1,j}^i, ..., W_{N,j}^i)\right\|_2 + \alpha\sum_{k=1}^{N}\sum_{i=1}^{d}\sum_{j=1}^{d}\left\|W_{k,j}^i\right\|_2. \tag{13}$$

We optimize the model architecture illustrated in Figure 2 using Eq.(13). Specifically, $H_{k,t}^i$ is obtained from the intervention identification network $g_{k,i}(X_{k,t};W_k^i)$, while $C_{k,t}^i$ is derived from the invariant Granger causal network $f_i(P_{k,t};W_0^i)$. These two representations are combined by the next-timestep embedding function $h_{k,i}(\cdot)$ to produce $E_{k,t+1}^i$, which is then projected to predict $X_{k,t+1}^i$ through the next-timestep prediction network. After optimization, non-zero entries in the observed-variable block of $f_i(\cdot)$ correspond to invariant causal relations to variable $i$ that are shared both across environments and over time, whereas $g_{k,i}(\cdot) = 0$ indicates that no incoming edge to variable $i$ is intervened in environment $k$.

## 4.3 IDENTIFIABILITY

**Theorem 1** (Identifiability of Granger Causal Graph). *Under the following assumptions:*

*(A1) The data are generated from the structural model in Eq.(2) across $N$ environments.*

*(A2) The Granger causal graph structure encoded by the support of $W_{0,1:d}$ is invariant across all environments. Modular interventions target only $X_t \rightarrow X_{t+1}$ edges.*

**(A3)** *Latent confounders are exogenous, i.e., $W_{0,d+1:d+p} = 0$, meaning past observed variables do not cause future latent changes. Moreover, the latent-to-observed mechanism $Z_t \to X_{t+1}$ is non-intervenable and its parameters are invariant across environments.*

**(A4)** *The model satisfies adjacency faithfulness, and the environments are sufficiently diverse in terms of interventions on observed variables and distributional shifts in the latent confounders (see Appendix A.1 for details).*

*Then, for each variable $X_{t+1}^i$, its Granger causal parent set $\mathrm{PA}(X_{t+1}^i) \subseteq \{X_t^1, \ldots, X_t^d\}$ is the unique minimal predictor set from the observed variables that remains invariant across all environments. At the population optimum of the objective in Eq.(12), the proposed model recovers the true Granger causal graph among the observed variables by restricting its discovered dependencies to that set:*

$$\{ j \in \{1, \ldots, d\} : \ \|(W_{0,j}^i, W_{1,j}^i, \ldots, W_{N,j}^i)\|_2 > 0 \} = \mathrm{PA}(X_{t+1}^i).$$

*Proof sketch.* Conditioning on $\mathrm{PA}(X_{t+1}^i)$ and $Z_t$ blocks all backdoor paths (Assumption 3), ensuring that the direct mechanism of $X_{t+1}^i$ remains invariant across environments (Assumption 2), so $\mathrm{PA}(X_{t+1}^i)$ is sufficient. If a true parent is excluded, Assumption 4 guarantees that invariance is violated; adding non-parents violates minimality, therefore $\mathrm{PA}(X_{t+1}^i)$ is the unique minimal invariant set (Peters et al., 2016). In Eq.(12), non-parents increase only the penalty while parents cannot be excluded without worsening risk, so the estimator selects precisely the true parent set $\mathrm{PA}(X_{t+1}^i)$. The temporal order $t \to t+1$ fixes edge directions and removes orientation ambiguity. $\square$

**Theorem 2** (Identifiability of the Latent Confounder Subspace). *With the same assumptions of Theorem 1, let $Z_t^\star \in \mathbb{R}^r$ be the ground-truth latent process, which has environment-invariant dynamics but environment-dependent marginal distributions. The data generation process for the observed variables is given by:*

$$X_{t+1} = f_X(W_{X_{t+1}X_t}^\top X_t + W_{X_{t+1}Z_t}^\top Z_t^\star) + \epsilon_{t+1}^X,$$

*where the influence of the latent variables is non-degenerate, i.e., the Jacobian $\frac{\partial f_X}{\partial z}$ has full rank $r$ on a set of positive measure. Let the latent process $Z_t \in \mathbb{R}^p$ learned by our model by minimizing the objective in Eq. (12) have dimension $p \geq r$. Then, at the population optimum, the learned latent subspace is equivalent to the ground-truth latent subspace up to an invertible linear transformation. That is, there exists an invertible matrix $R \in \mathbb{R}^{r \times r}$ such that:*

$$\mathrm{span}(Z_t) = \mathrm{span}(Z_t^\star).$$

*Proof sketch.* Since the Jacobian $\frac{\partial f_X}{\partial z}$ has full column rank $r$, each latent factor has a linearly independent effect on $X_{t+1}$. Thus, if a true factor is not included, environment–induced variation along that direction cannot be absorbed by the retained variables; the conditional law of $X_{t+1}$ then depends on the environment, which violates invariance. Conversely, adding latent variables beyond the true ones does not reduce population risk but increases the regularization term, so their coefficients are driven to zero at the optimum. Consequently, the learned $Z_t$ spans the same subspace as the ground-truth $Z_t^\star$, up to an invertible linear transformation (Hyvarinen et al., 2019; Khemakhem et al., 2020). $\square$

**Theorem 3** (Identifiability of Edge-level Interventions). *With the same assumptions of Theorem 1, and given that the latent confounder subspace is correctly recovered as established in Theorem 2. Let $W_{0,1:d}$ denote the invariant $X_t \to X_{t+1}$ mechanism and, for each environment $k$, let $W_k$ denote the environment-specific deviation on $X_{k,t} \to X_{k,t+1}$ (the $Z_t \to X_{t+1}$ mechanism has no deviation by Assumption 3), and assume that these deviations are sparse across environments. Then, for any observed edge $j \to i$ and any environment $k$,*

$$W_{k,j}^i \neq 0 \quad \Longleftrightarrow \quad \text{the edge } j \to i \text{ is intervened in environment } k.$$

*Proof sketch.* The mechanism among observed variables in environment $k$ is decomposed into an invariant part $W_{0,1:d}$ and an environment-specific deviation $W_k$. If edge $j \to i$ is not intervened in $k$, its true parameter equals the invariant one, so the risk is minimized by $W_{k,j}^i = 0$; any nonzero deviation fails to decrease the risk and contributes only to the regularization term. If the edge

is intervened, its true parameter differs from the invariant value. Under Assumption 2-4 and the parameterization in Eq.(12), the discrepancy on $j \to i$ cannot be absorbed elsewhere; therefore the risk–minimizing solution requires $W_{k,j}^i \neq 0$. $\square$

**Corollary 1** (Identifiability of Node-level Interventions). *Under the setting of Theorem 3, a node-level intervention on variable $X_{t+1}^i$ in environment $k$ is identifiable. Specifically,*

$$\{ X_t^j : W_{k,j}^i \neq 0 \} \ = \ \mathrm{PA}(X_{t+1}^i) \quad \Longleftrightarrow \quad \textit{node } i \textit{ is intervened in environment } k,$$

## 5 EXPERIMENTS

In this paper, we compare our InvarGC to state-of-the-art (SOTA) baselines, including GC (Granger, 1969), NGC (Tank et al., 2021), eSRU (Khanna & Tan, 2019), CUTS (Cheng et al., 2023; 2024), DyNOTEARS (Pamfil et al., 2020), GVAR (Marcinkevičs & Vogt, 2020), JRNGC (Zhou et al., 2024), and LPCMCI (Gerhardus & Runge, 2020).

We take two established metrics: Area Under the Receiver Operating Characteristic Curve (AUROC) and Area Under the Precision-Recall Curve (AUPRC). An AUROC value of 0.5 or lower signifies poor performance. In contexts where causal relationships are sparse, AUPRC serves as a more reliable measure of a model's effectiveness in identifying causal relationships. This reliability is because its focus on the accurate detection of positive cases, which is crucial in situations with a limited number of true causal relationships.

### 5.1 EXPERIMENTAL SETUP

We begin by briefly introducing the process used to generate the synthetic data, which is based on the functional causal model (Huang et al., 2020) described in Eq.(14).

$$X_{t+1}^i = \sum_{X_t^j \in \mathrm{PA}(X_{t+1}^i)} f_{ij}(X_t^j) + \epsilon_{t+1}^i, \tag{14}$$

where $f_{ij}(\cdot)$ may be specified as a linear, cubic, $\tanh$, or sin function, or alternatively as a multi-layer perceptron (MLP) network with randomly initialized weights. The noise term $\epsilon_{i,t+1}$ is generated from a standard normal distribution $\mathcal{N}(0,1)$.

**Synthetic Time Series Data.** We generate synthetic multivariate time series data to evaluate and benchmark causal discovery performance under latent confounding and unknown interventions. The ground-truth causal graph consists of $d = 5$ observed variables and $p = 1$ latent confounder. The Granger summary graph over the observed variables is randomly generated with an edge probability of $e = 0.3$. The latent confounder serves as a common cause to two randomly selected observed variables. The time series are generated following a first-order vector autoregressive process, where the state of the $d$ observed variables at time $t + 1$ is determined by the state of all $d + p$ variables at $t$. To introduce non-linearity, we apply a Leaky ReLU activation with a negative slope of $0.01$. To simulate a realistic discovery setting, we generate data for three environments: two remain purely observational, while one is subject to edge-level interventions (i.e., modifying the corresponding coefficients in the parameter matrix). Importantly, all competing methods have no information about which environment is intervened.

**Real-world Time Series Data.** We also evaluate all competing causal discovery methods on the real-world benchmarks, including **1) Tennessee Eastman Process (TEP) dataset** (Downs & Vogel, 1993), a widely used benchmark in time series anomaly detection with ground-truth causal graphs. The dataset contains 33 variables, 960 observations, and 21 predefined anomalies, each anomaly representing a distinct environment. We construct two versions based on TEP data; *Conf-TEP (w/o Interventions)* uses only the normal time series, where a variable without parents but with multiple children is masked as a latent confounder. *Conf-TEP* further incorporates intervention-induced anomalies across multiple environments. We further evaluate our approach on **2) the Causal-Rivers dataset** (Stein et al., 2025), a large-scale time series causal discovery benchmark constructed from river discharge measurements across Germany. We generated four representative subsets of the Causal-Rivers dataset: *Random*, comprising diverse connected subgraphs that naturally include a mixture of intervention-like effects and confounding patterns; *Confounder*, which simulates latent

confounding by removing a parent node to introduce an unobserved common cause; *Flood*, corresponding to periods of extreme rainfall with strong non-stationarity involving 42 nodes in the RiversElbeFlood region; and *No Rain + Flood*, which concatenates a stationary no-rain period and a non-stationary flood period simulate both non-intervened (no-rain) and intervened (flood) environments.

## 5.2 Experimental Results and Analysis

The results in Table 1 indicate that InvarGC consistently achieves the highest AUROC and AUPRC scores, attaining perfect recovery in both linear and non-linear synthetic datasets. This is because the synthetic data, though containing both latent confounders and unknown interventions, remain consistent with the invariance assumptions of InvarGC, enabling precise separation between causal and non-causal edges. While some deep learning–based methods also perform strongly on synthetic settings, their effectiveness is undermined when both confounders and interventions are introduced, as they are not designed to jointly address these challenges. On the more challenging *Conf-TEP w/o Interventions* datasets, all methods experience performance degradation, reflecting the inherent difficulty of causal discovery under hidden confounding and sparse causal structures. Although some baselines achieve relatively high AUROC scores, their low AUPRC values reveal limited ability to recover true causal edges in such sparse settings. Interventions in *Conf-TEP* dataset provide partial improvements by helping distinguish consistent relationships from spurious ones, yet most methods remain ill-suited for the combined challenges of interventions, latent confounding, and heterogeneous subsystem dynamics. Explicitly accounting for such complexity enables InvarGC to discover more reliable causal relationships and consistently outperform all baselines.

Table 1: Performance Evaluation on Synthetic and TEP Data. **Red**: the best, Blue: the 2nd best.

| Methods | Linear | | Non-Linear | | Conf-TEP (w/o I) | | Conf-TEP | |
|---|---|---|---|---|---|---|---|---|
| | AUROC | AUPRC | AUROC | AUPRC | AUROC | AUPRC | AUROC | AUPRC |
| GC | 0.6667 | 0.3684 | 0.6111 | 0.3333 | 0.6524 | 0.0877 | 0.6215 | 0.0795 |
| CD-NOD | 0.8611 | 0.4167 | 0.8056 | 0.2917 | 0.6046 | 0.1185 | 0.6096 | 0.0697 |
| LPCMCI | 0.7540 | 0.5387 | 0.6032 | 0.4448 | 0.6214 | 0.0821 | 0.6735 | 0.0829 |
| NGC | 0.8968 | 0.8187 | 0.6349 | 0.6063 | 0.6338 | 0.3140 | 0.6773 | 0.4100 |
| eSRU | 0.9365 | 0.8333 | 0.9841 | 0.9617 | 0.6656 | 0.2246 | 0.7503 | 0.4074 |
| CUTS | 0.9127 | 0.7997 | 0.8968 | 0.7742 | 0.6716 | 0.1864 | 0.6963 | 0.2884 |
| DyNotears | 0.7817 | 0.5962 | 0.7579 | 0.5925 | 0.6883 | 0.2743 | 0.7022 | 0.3359 |
| GVAR | 0.9206 | 0.8072 | 0.8016 | 0.7087 | 0.6733 | 0.2510 | 0.7112 | 0.2679 |
| JRNGC | 0.7857 | 0.6445 | 0.8254 | 0.6828 | 0.6098 | 0.1877 | 0.6627 | 0.2620 |
| **InvarGC** | **1.0000** | **1.0000** | **1.0000** | **1.0000** | **0.7855** | **0.3567** | **0.8121** | **0.4380** |

The results in Table 2 show that InvarGC achieves the best overall performance across subsets of the Causal-Rivers benchmark, consistently ranking among the top two in both AUROC and AUPRC. In the *Random* subset, its performance is comparable to the strongest baselines, whereas in the *Confounder* subset it clearly outperforms all competitors, underscoring its robustness to hidden confounding. In the *Flood* and *No Rain + Flood* subsets, which correspond to highly non-stationary and heterogeneous environments, InvarGC again delivers the most reliable performance. These findings confirm that invariance-based modeling is well suited to handle real-world complexities such as interventions, latent confounders, and distributional shifts. While several methods remain competitive in the *Random* subset, the *Confounder* subset increases the effect of hidden confounding, causing most methods to degrade. Moreover, results from the *Flood* and *No Rain + Flood* subsets indicate that some methods can take advantage of pronounced distributional shifts, since such variability offers valuable information for uncovering causal structure.

## 5.3 Ablation Study

In this section, we present an ablation study from three perspectives. First, we evaluate the contribution of the latent confounder inference module (LCIM). Second, we conduct a sensitivity analysis of $L$ in Eq.(13) with respect to the true number of latent confounders $|Z|$, and examine the consequences when $|Z| > L$. Finally, we assess the impact of the intervention identification network.

Table 2: Performance Evaluation on Causal-Rivers Data. **Red**: the best, Blue: the 2nd best.

| METHODS | RANDOM | | CONFOUNDER | | FLOOD | | NORAIN + FLOOD | |
|---|---|---|---|---|---|---|---|---|
| | AUROC | AUPRC | AUROC | AUPRC | AUROC | AUPRC | AUROC | AUPRC |
| GC | 0.5313 | 0.3750 | 0.5079 | 0.4415 | 0.7006 | 0.0847 | 0.7122 | 0.0940 |
| CD-NOD | 0.5938 | 0.3432 | 0.5741 | 0.2778 | 0.5073 | 0.0368 | 0.4782 | 0.0241 |
| LPCMCI | 0.5625 | 0.4381 | 0.5714 | 0.6095 | 0.7683 | 0.5631 | 0.6954 | 0.5456 |
| NGC | 0.6042 | 0.5156 | 0.4762 | 0.5441 | 0.7747 | **0.5814** | 0.7627 | 0.5736 |
| eSRU | 0.7778 | 0.7671 | 0.8571 | 0.8429 | 0.6925 | 0.3606 | 0.7627 | 0.3435 |
| CUTS | 0.8194 | 0.8267 | 0.8254 | 0.8691 | 0.7247 | 0.3362 | 0.7513 | 0.4513 |
| DYNOTEARS | 0.8056 | 0.8313 | 0.7143 | 0.6301 | 0.7623 | 0.5599 | 0.7796 | 0.5701 |
| GVAR | 0.9028 | 0.8817 | 0.6825 | 0.6372 | 0.6602 | 0.2949 | 0.6892 | 0.2983 |
| JRNGC | 0.8958 | **0.9062** | 0.6508 | 0.6810 | 0.7414 | 0.3914 | 0.7062 | 0.2508 |
| **INVARGC** | **0.9097** | 0.9038 | **0.9048** | **0.9341** | **0.7788** | 0.5750 | **0.7832** | **0.5839** |

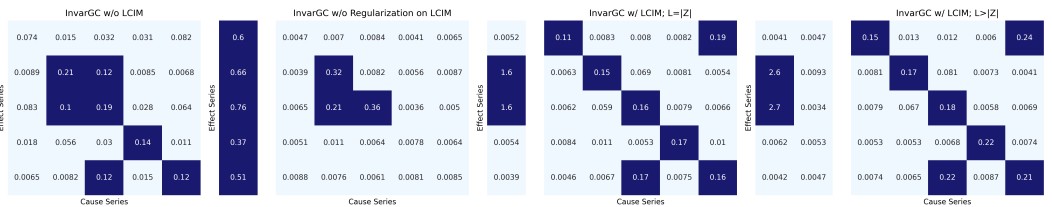

Figure 3: Ablation study of the latent confounder inference module (LCIM). We compare four settings: (i) w/o LCIM, (ii) w/ LCIM but without regularization, (iii) w/ LCIM with $L$ set to the true number of latent confounders ($L = |Z|$), and (iv) w/ LCIM with $L > |Z|$.

We first evaluate the effectiveness of LCIM. Without explicitly modeling latent confounders, the estimated causal structure contains spurious correlations. With the regularization term in Eq. (13), LCIM achieves substantially better performance by effectively absorbing the influence of confounders, therefore reducing spurious associations and enabling more accurate recovery of the true Granger causal graph. We also conduct a sensitivity analysis on the parameter $L$. Since the true number of confounders $Z$ is unknown in real-world applications, we examine the cases where $L \geq |Z|$, with a properly chosen $\lambda_z$, the model can still accurately recover both the latent confounders and the underlying causal structures, as illustrated in Figure 3. On the other hand, when $|Z| > L$, the model can only partially absorb the confounding effects, leaving residual spurious correlations and resulting in biased causal graph estimation, which is consistent with the theoretical limits of identifiability under model misspecification.

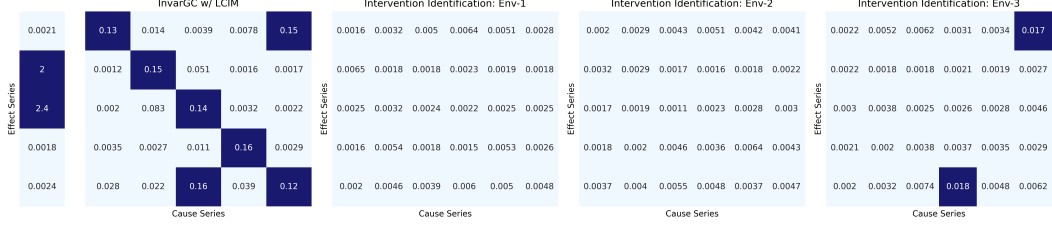

Figure 4: Recovered Granger causality and intervention identification across three environments.

We finally evaluate edge-level intervention identification within each environment in the presence of latent confounders and without access to intervention labels (Figure 4). We observe that the method tends to prioritize strong interventions while overlooking milder ones. In our experiments, we set the threshold based on the order of magnitude of the model weights. We also proposed proximal gradient descent algorithm with group soft-thresholding in Appendix A.4, which enforces sparsity by driving entire parameter groups to exactly zero.

## 6 CONCLUSION

In this paper, we introduce InvarGC, a method for inferring invariant Granger causal graph from heterogeneous interventional time series data under latent confounding even when interventional targets are unknown. We further establish identifiability of the proposed approach under appropriate assumptions. Experiments on synthetic and real-world datasets demonstrate its effectiveness. For future work, we are going to leverage the learned causal graphs in real-world applications such as domain adaptation, anomaly detection, and time series forecasting.

## 7 ETHICS STATEMENT

The authors acknowledge the use of large language models (LLMs) for the limited purpose of grammar checking and language polishing. No LLMs were used for data analysis, methodological design, or generation of scientific content. All ideas, results, and conclusions presented in this manuscript are the full responsibility of the authors. This work uses only synthetic and publicly available datasets, without involving human subjects or sensitive personal data. We do not foresee any ethical concerns arising from this study.

## 8 REPRODUCIBILITY STATEMENT

The formal problem setup and assumptions are presented in Section 4, with complete theoretical results and proofs provided in Appendix A.1. The experimental settings with dataset descriptions, preprocessing steps, hyperparameter configurations and evaluation protocol are provided in Section 5.1 and A.2. We include the complete source code and bash scripts in the supplementary materials, with all training parameters specified to enable exact replication of our experiments. All datasets used are publicly available, and references to their sources are included in the main text.

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

## A APPENDIX

### A.1 PROOFS AND DISCUSSION

This appendix provides the detailed proofs for the identifiability theorems presented in the Section 4.3. We first establish two key lemmas and then use them to prove the main results for the identifiability of the Granger causal graph, latent confounder subspace, and interventions. We also discussed the limitations of our theoretical analysis and directions for future work.

**Lemma 1** (Blocking Backdoor Paths). *Under Assumption 3, conditioning on the true parent set* $\mathrm{PA}(X_{t+1}^i)$ *and the true latent confounders* $Z_t^\star$ *blocks all backdoor paths into* $X_{t+1}^i$.

*Proof.* Assumption 3 ensures that there are no causal paths of the form $X \to Z^\star$. Any remaining backdoor from $X_t^j$ to $X_{t+1}^i$ must then pass through $Z_t^\star$, so conditioning on $\mathrm{PA}(X_{t+1}^i)$ and $Z_t^\star$ blocks all backdoor paths into $X_{t+1}^i$. $\square$

**Lemma 2** (Uniqueness of Minimal Invariant Set). *Under Assumption 2-4, the unique minimal set of observed variables that renders the conditional distribution of $X_{t+1}^i$ invariant across environments (when conditioned together with $Z_t^\star$) is the true parent set $\mathrm{PA}(X_{t+1}^i)$.*

*Proof. (Sufficiency)* By Lemma 1, conditioning on $\mathrm{PA}(X_{t+1}^i)$ and $Z_t^\star$ blocks all backdoor paths. The remaining direct causal mechanism from $Z_t^\star$ to $X_{t+1}^i$ is invariant by Assumption 3. The full set of parents is thus sufficient for invariance by Assumption 2.

*(Minimality)* If a true parent $X_t^j \in \mathrm{PA}(X_{t+1}^i)$ is omitted, Assumption 4 ensures the environments provide sufficiently diverse interventions such that there exists at least one environment in which the mechanism of the edge $j \to i$ differs from its invariant form. In that environment, the conditional distribution of $X_{t+1}^i$ given the reduced set of variables changes relative to other environments, thereby violating invariance. Conversely, if a non-parent variable is included, it does not contribute to predicting $X_{t+1}^i$ once the true parents are already conditioned on, so the set fails to be minimal. Hence the observed parent set $\mathrm{PA}(X_{t+1}^i)$ is the unique minimal invariant predictor set among observed variables, in line with the principle of Invariant Causal Prediction (Peters et al., 2016). $\square$

**Theorem 1** (Identifiability of Granger Causal Graph). *Under the following assumptions:*

*(A1) The data are generated from the structural model in Eq.(2) across $N$ environments.*

*(A2) The Granger causal graph structure encoded by the support of $W_{0,1:d}$ is invariant across all environments. Modular interventions target only $X_t \to X_{t+1}$ edges.*

*(A3) Latent confounders are exogenous, i.e., $W_{0,d+1:d+p} = 0$, meaning past observed variables do not cause future latent changes. Moreover, the latent-to-observed mechanism $Z_t \to X_{t+1}$ is non-intervenable and its parameters are invariant across environments.*

*(A4) The model satisfies adjacency faithfulness, and the environments are sufficiently diverse in terms of interventions on observed variables and distributional shifts in the latent confounders.*

*Then, for each variable $X_{t+1}^i$, its Granger causal parent set $\mathrm{PA}(X_{t+1}^i) \subseteq \{X_t^1, \ldots, X_t^d\}$ is the unique minimal predictor set from the observed variables that remains invariant across all environments. At the population optimum of the objective in Eq.(12), the proposed model recovers the true Granger causal graph among the observed variables by restricting its discovered dependencies to that set:*

$$\{ j \in \{1, \ldots, d\} : \ \|(W_{0,j}^i, W_{1,j}^i, \ldots, W_{N,j}^i)\|_2 > 0 \} = \mathrm{PA}(X_{t+1}^i).$$

*Proof.* Consider a fixed target $X_{t+1}^i$, for each candidate predictor $X_t^j \in \{X_t^1, \ldots, X_t^d\}$, write the $(N+1)$-dimensional group $\mathcal{G}_{i,j} := (W_{0,j}^i, \ldots, W_{N,j}^i)$. Let $\mathcal{L}_i(W, Z)$ denote the population objective in Eq. (12) restricted to the $i$-th equation, i.e.,

$$\mathcal{L}_i(W, Z) = \mathcal{R}_i(W, Z) + (1 - \alpha) \sum_{j=1}^{d+p} \|\mathcal{G}_{i,j}\|_2 + \alpha \sum_{k=1}^{N} \sum_{j=1}^{d} \|W_{k,j}^i\|_2 + \lambda_z \, \mathcal{R}_Z(Z),$$

where $\mathcal{R}_i(W, Z)$ is the population squared risk for predicting $X_{t+1}^i$, and

$$\mathcal{R}_Z(Z) := \sum_{k=1}^{N} \sum_{l=1}^{L} \sqrt{\frac{1}{T} \sum_{t=1}^{T} Z_{k,l,t}^2}$$

is the latent confounder regularization term.

*(i) Non-parents are excluded.* Take any non-parent $X_t^j \notin \mathrm{PA}(X_{t+1}^i)$. By Lemma 2 and Assumptions 2–4, the population conditional mean of $X_{t+1}^i$ given the true conditioning set $\{\mathrm{PA}(X_{t+1}^i), Z_t^\star\}$ does not depend on $X_t^j$ and is invariant across environments. Equivalently, in the population normal equations for the linear objective, the cross-covariance between the residual (after projecting onto

the true set) and $X_t^j$ is zero in every environment, so the unique risk minimizer in the $j$-th direction is $\mathcal{G}_{i,j} = 0$:

$$\inf_{v \in \mathbb{R}^{N+1}} \mathcal{R}_i\big(\mathcal{G}_{i,j} = v\big) = \mathcal{R}_i\big(\mathcal{G}_{i,j} = 0\big), \quad \text{with equality iff } v = 0.$$

Adding any $v \neq 0$ cannot reduce $\mathcal{R}_i$ and strictly increases the group penalty terms $(1-\alpha)\|v\|_2 + \alpha \sum_k \|v_k\|_2$. Hence for any non-parent,

$$\mathcal{L}_i\big(\mathcal{G}_{i,j} = v\big) > \mathcal{L}_i\big(\mathcal{G}_{i,j} = 0\big) \quad (v \neq 0),$$

and the population minimizer sets the entire group to zero.

*(ii) True parents are retained.* Now take any true parent $X_t^j \in \text{PA}(X_{t+1}^i)$. By Lemma 2 and Assumption 4, there exists at least one environment in which the mechanism on the edge $j \to i$ differs from the invariant value. If we force $\mathcal{G}_{i,j} = 0$ (equivalently, omit $X_t^j$), the best achievable population risk is strictly larger than the oracle risk attained when $\mathcal{G}_{i,j}$ is free:

$$\Delta_{i,j} := \inf_{W : \mathcal{G}_{i,j}=0} \mathcal{R}_i(W, Z^\star) - \inf_W \mathcal{R}_i(W, Z^\star) > 0.$$

Let $\mathcal{G}_{i,j}^\star$ be any population risk minimizer without the restriction $\mathcal{G}_{i,j} = 0$. For regularization parameters in a standard non-degenerate range (i.e., small enough that $(1-\alpha)\|\mathcal{G}_{i,j}^\star\|_2 + \alpha \sum_k \|W_{k,j}^{i\star}\|_2 < \Delta_{i,j}$), we have

$$\mathcal{L}_i(\mathcal{G}_{i,j} = 0, Z^\star) = \inf_{W : \mathcal{G}_{i,j}=0} \mathcal{R}_i(W, Z^\star)$$

$$+ (1-\alpha)\sum_{\ell \neq j}^{d+p} \|\mathcal{G}_{i,\ell}\|_2 + \alpha \sum_{k=1}^N \sum_{\ell \neq j}^d \|W_{k,\ell}^i\|_2 + \lambda_z \mathcal{R}_Z(Z^\star)$$

$$> \mathcal{R}_i(W^\star, Z^\star)$$

$$+ (1-\alpha)\sum_{\ell=1}^{d+p} \|\mathcal{G}_{i,\ell}^\star\|_2 + \alpha \sum_{k=1}^N \sum_{\ell=1}^d \|W_{k,\ell}^{i\star}\|_2 + \lambda_z \mathcal{R}_Z(Z^\star)$$

$$= \mathcal{L}_i(W^\star, Z^\star).$$

so zeroing the entire group increases the objective. Therefore the optimizer must retain $\mathcal{G}_{i,j} \neq 0$ for every true parent.

Combining (i) and (ii), the set of indices with nonzero groups satisfies

$$\big\{ X_t^j : \|\mathcal{G}_{i,j}\|_2 > 0 \big\} = \text{PA}(X_{t+1}^i),$$

which proves the claim. $\qquad\square$

In Assumption 4, "sufficiently diverse" has two components: (i) sufficiently diverse interventions on the observed variables, and (ii) sufficiently diverse distributional shifts in the latent confounders across environments. First, for each target variable $X_{t+1}^i$ and each environment $k$, the environment-specific deviation vector $W_k^i \in \mathbb{R}^d$ encodes which incoming edges $X_t^j \to X_{t+1}^i$ are modified by interventions. From $\{W_k^i\}_{k=1}^N$, we can derive an intervention signature matrix for target $i$: its $(k, j)$-entry is 1 if the edge $X_t^j \to X_{t+1}^i$ is intervened in environment $k$, and 0 otherwise. The column corresponding to edge $j$ therefore records in which environments this edge is intervened. If the entire column is zero for a true parent edge, that edge is never intervened in any environment, making it impossible to separate its invariant causal effect from spurious associations at the population level. We say that interventions on the observed variables are sufficiently diverse if, for each target $X_{t+1}^i$: (a) every true parent edge $X_t^j \to X_{t+1}^i$ is intervened in at least one environment (its column is not identically zero), and (b) the intervention signature matrix restricted to the true parent columns has full column rank, so that the intervention patterns of different parents are not redundant. Intuitively, (b) rules out degenerate cases where parents are always intervened in exactly the same way and hence cannot be disentangled. Second, let $k \in \{1, \ldots, N\}$ index environments and let $P_k(Z_t)$ denote the marginal distributions of the latent confounders $Z_t$ in environment $k$. We assume that the latent-to-observed mechanism is invariant across environments (as in Assumption 3), i.e.,

$$X_{t+1} = f_X\big(W_{X_{t+1}X_t}^\top X_t + W_{X_{t+1}Z_t}^\top Z_t\big) + \epsilon_{t+1}^X,$$

with the same $f_X$, $W_{X_{t+1}X_t}$, and $W_{X_{t+1}Z_t}$ for all $k$, while the marginal distribution of $Z_t$ is allowed to vary with $k$. Formally, write $\mu_k := \mathbb{E}_k[Z_t]$ and $\Sigma_k := \mathrm{Cov}_k(Z_t)$ for the mean and covariance of $Z_t$ in environment $k$. We say that the latent distributions are sufficiently diverse if the collection $\{P_k(Z_t)\}_{k=1}^N$ induces linearly independent variations in the effective confounding term $W_{X_{t+1}Z_t}^\top Z_t$ across environments. A simple sufficient condition is that the set of projected means $\{W_{X_{t+1}Z_t}^\top \mu_k\}_{k=1}^N$ span the same subspace as $W_{X_{t+1}Z_t}^\top Z_t^\star$, i.e.,

$$\mathrm{span}\big( W_{X_{t+1}Z_t}^\top \mu_1, \ldots, W_{X_{t+1}Z_t}^\top \mu_N \big) \;=\; \mathrm{span}\big( W_{X_{t+1}Z_t}^\top Z_t^\star \big).$$

Intuitively, different environments must induce non-redundant shifts in the confounders along directions that actually affect $X_{t+1}$. Otherwise, environment-induced variation in $Z_t$ could be absorbed into the noise and the latent subspace would not be identifiable from $X$ alone.

**Theorem 2** (Identifiability of the Latent Confounder Subspace). *With the same assumptions of Theorem 1, let $Z_t^\star \in \mathbb{R}^r$ be the ground-truth latent process, which has environment-invariant dynamics but environment-dependent marginal distributions. The data generation process for the observed variables is given by:*

$$X_{t+1} = f_X(W_{X_{t+1}X_t}^\top X_t + W_{X_{t+1}Z_t}^\top Z_t^\star) + \epsilon_{t+1}^X,$$

*where the influence of the latent variables is non-degenerate, i.e., the Jacobian $\frac{\partial f_X}{\partial z}$ has full rank $r$ on a set of positive measure. Let the latent process $Z_t \in \mathbb{R}^p$ learned by our model by minimizing the objective in Eq. (12) have dimension $p \geq r$. Then, at the population optimum, the learned latent subspace is equivalent to the ground-truth latent subspace up to an invertible linear transformation. That is, there exists an invertible matrix $R \in \mathbb{R}^{r \times r}$ such that:*

$$\mathrm{span}(Z_t) = \mathrm{span}(Z_t^\star).$$

*Proof.* The proof rests on three points. *(i) Equivalence under invertible reparametrization.* For any invertible $R \in \mathbb{R}^{r \times r}$, the reparametrization $Z_t^\star \mapsto Z_t'^\star := R^\top Z_t^\star$ and $W_{X_{t+1}Z_t} \mapsto W'_{X_{t+1}Z_t} := R^{-1}W_{X_{t+1}Z_t}$ yields the same conditional mean,

$$f_X\big(W_{X_{t+1}X_t}^\top X_t + W'^\top_{X_{t+1}Z_t} Z_t'^\star\big) = f_X\big(W_{X_{t+1}X_t}^\top X_t + W_{X_{t+1}Z_t}^\top Z_t^\star\big).$$

Hence only the latent subspace $\mathrm{span}(Z_t^\star)$ is identifiable.

*(ii) Extraneous latent directions are eliminated.* Write any learned $Z_t \in \mathbb{R}^p$ as

$$Z_t \;=\; A^\top Z_t^\star + U_t, \qquad A \in \mathbb{R}^{r \times p}, \quad U_t \perp \mathrm{span}(Z_t^\star).$$

Let $\mathcal{L}(W, Z)$ be the population objective in Eq. (12) and $\mathcal{R}_Z(Z)$ the latent regularizer already defined in the text. For any fixed $i$, denote by $\mathcal{R}_i(W, Z)$ the population squared risk for predicting $X_{t+1}^i$. Because $U_t$ lies outside $\mathrm{span}(Z_t^\star)$ and $\frac{\partial f_X}{\partial z}$ has full column rank $r$, the population normal equations (after conditioning on $\{X_t, Z_t^\star\}$ per Lemma 1) imply that the cross-covariance between the residual and $U_t$ is zero in every environment. Therefore, along any coefficient direction attached to $U_t$, the risk $\mathcal{R}_i$ is uniquely minimized at zero. Adding any nonzero weight on $U_t$ cannot reduce $\mathcal{R}_i$ and strictly increases the penalty $\lambda_z \mathcal{R}_Z(Z)$ (and, if parameterized, the corresponding weight penalties). Thus, at the population optimum, the coefficients on $U_t$ must vanish, and the effective learned latent variables lie in $\mathrm{span}(Z_t^\star)$. Formally, for each target $i$ and any coefficient block $b$ attached to $U_t$,

$$\inf_v \mathcal{R}_i(\mathrm{coeff}(U_t) = v) \;=\; \mathcal{R}_i(\mathrm{coeff}(U_t) = 0), \quad \text{with equality iff } v = 0,$$

so $\mathcal{L}$ is strictly larger whenever $v \neq 0$. Hence extraneous directions are eliminated at the optimum.

*(iii) Missing true latent directions strictly increase risk.* Suppose, towards a contradiction, that the learned latent subspace has dimension $m < r$ (or, more generally, that $\mathrm{span}(Z_t)$ fails to contain $\mathrm{span}(Z_t^\star)$). Then there exists a nonzero direction $a \in \mathbb{R}^r$ such that $a^\top Z_t^\star$ is orthogonal to $\mathrm{span}(Z_t)$. Because $\frac{\partial f_X}{\partial z}$ has full column rank $r$, variation along $a^\top Z_t^\star$ induces a nondegenerate change in the conditional mean of $X_{t+1}$, and by Assumption 4 there exists an environment in which the mechanism along that direction differs from the invariant one. Since $a^\top Z_t^\star$ is not representable within $\mathrm{span}(Z_t)$, the resulting environment-induced variation cannot be absorbed by the model;

hence the conditional law of $X_{t+1}$ given $(X_t, Z_t)$ varies across environments, violating invariance and yielding a strictly larger population risk than the oracle that uses the full $Z_t^\star$:

$$\Delta := \inf_W \mathcal{R}(W, Z) - \inf_W \mathcal{R}(W, Z^\star) > 0.$$

For regularization parameters in a standard non-degenerate range, the increase $\Delta$ dominates any penalty saving, so the population optimum cannot occur with $m < r$ (or with $\mathrm{span}(Z_t)$ missing part of $\mathrm{span}(Z_t^\star)$).

Combining (ii) and (iii), at any population optimum with $p \geq r$ the learned latent variables $Z_t$ span exactly $\mathrm{span}(Z_t^\star)$; together with (i) this proves subspace identifiability up to an invertible linear transformation. $\qquad\square$

**Theorem 3** (Identifiability of Edge-Level Interventions). *With the same assumptions of Theorem 1, and given that the latent confounder subspace is correctly recovered as established in Theorem 2. Let $W_{0,1:d}$ denote the invariant $X_t \rightarrow X_{t+1}$ mechanism and, for each environment $k$, let $W_k$ denote the environment-specific deviation on $X_{k,t} \rightarrow X_{k,t+1}$ (the $Z_t \rightarrow X_{t+1}$ mechanism has no deviation by Assumption 3), and assume that these deviations are sparse across environments. Then, for any observed edge $j \rightarrow i$ and any environment $k$,*

$$W_{k,j}^i \neq 0 \quad \Longleftrightarrow \quad \text{the edge } j \rightarrow i \text{ is intervened in environment } k.$$

*Proof.* For each environment $k$, the total mechanism on edge $j \rightarrow i$ is

$$\theta_{k,j}^i = W_{0,j}^i + W_{k,j}^i,$$

where $W_{0,j}^i$ is the invariant coefficient and $W_{k,j}^i$ is the environment-specific deviation.

*($\Rightarrow$) If the edge is intervened.* By Assumption 2, an intervention on $j \rightarrow i$ in environment $k$ implies that the true coefficient $\theta_{k,j}^{i,\mathrm{true}}$ differs from the invariant value $W_{0,j}^i$. Because the invariant mechanism and other edges cannot absorb this discrepancy (faithfulness, Assumption 4), the risk-minimizing solution must set $W_{k,j}^i \neq 0$ to match the true mechanism.

*($\Leftarrow$) If the edge is not intervened.* Then the true coefficient equals the invariant value $W_{0,j}^i$. Any solution with $W_{k,j}^i \neq 0$ necessarily mis-specifies the coefficient, increasing the population risk, and also incurs a positive penalty term $\alpha \|W_{k,j}^i\|_2$. Thus the optimal solution satisfies $W_{k,j}^i = 0$. $\qquad\square$

In Theorem 3 we implicitly assume that interventions are sparse, i.e., the invariant mechanism is the majority pattern and interventions occur only in a minority of environments. Concretely, for each edge $X_t^j \rightarrow X_{t+1}^i$, let $W_{k,j}^i$ denote its deviation parameter in environment $k$. We assume that the index set

$$S_{i,j} := \{ k \in \{1, \ldots, N\} : W_{k,j}^i \neq 0 \}$$

satisfies $|S_{i,j}| < N/2$.

**Corollary 1** (Identifiability of Node-level Interventions). *Under the setting of Theorem 3, a node-level intervention on variable $X_{t+1}^i$ in environment $k$ is identifiable. Specifically,*

$$\{ X_t^j : W_{k,j}^i \neq 0 \} = \mathrm{PA}(X_{t+1}^i) \quad \Longleftrightarrow \quad \text{node } i \text{ is intervened in environment } k,$$

*Proof.* Under the setting of Theorem 3, a node-level intervention on $X_{t+1}^i$ in environment $k$ is defined as an intervention on *all* of its incoming parent edges. Equivalently, this means that for every $X_t^j \in \mathrm{PA}(X_{t+1}^i)$ the deviation parameter $W_{k,j}^i \neq 0$. Therefore, node-level interventions are identifiable under this setting. $\qquad\square$

Our identifiability results are established in a linear setting and with a single time lag ($t \rightarrow t+1$). This is justified by the assumption that $X_t$ already summarizes the relevant history, so one-step dynamics suffice for Granger causal identification. The InvarGC model employs neural networks to capture non-linear mechanisms. Our theoretical analysis thus provides a foundation for extending this framework to more general non-linear settings.

## A.2 Dataset Construction Details

As illustrated in Figure 5, during the synthetic time series data generation process we masked a variable that has no parents but influences two other observed variables, treating it as a latent confounder. The latent structure created by this masking is highlighted with a red rectangle. This induces spurious correlations in the observational data, shown with a red background, while the remaining blue without blurring variables are treated as ground truth. We further simulated unknown interventions by randomly perturbing edge weights, which are blurred in blue.

For the real-world Tennessee Eastman Process (TEP) dataset, we followed the same assumptions and applied the masking strategy to construct a Confounded TEP (Conf-TEP) dataset.

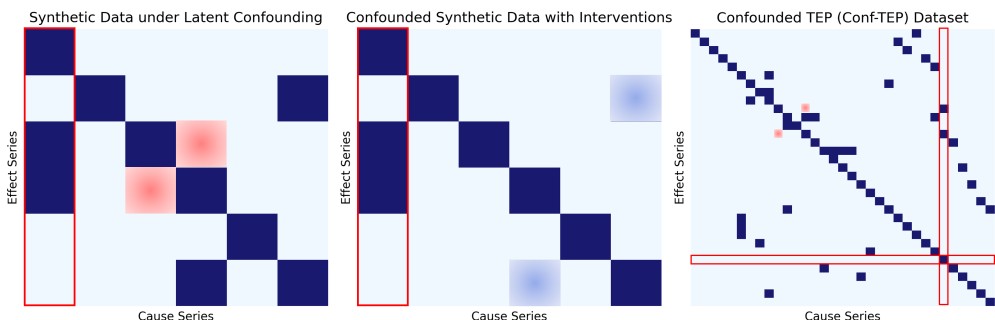

Figure 5: Illustration of latent confounding and interventions in synthetic and Conf-TEP data.

To construct the confounded dataset from the Causal-Rivers data, we selected six subsets whose causal graphs are identical, consistent with the assumptions made in our study. Each subset contains five observable variables. Following our masking strategy, we masked one variable that has no parents but influences multiple children, thereby introducing latent confounding. For the *Flood* dataset, we extracted 1,000 consecutive data points from the period 2024-09-09 to 2024-10-10. In addition, we used two equal-length windows from the preceding no-rain period to construct a mixed *NoRain+Flood* dataset. In the evaluation process, we use AUROC and AUPRC, for both metrics, the input consists of a one-dimensional vector of predicted edge scores obtained by flattening the edge-score matrix and a matching one-dimensional binary label vector obtained by flattening the ground-truth adjacency.

## A.3 Time Complexity and Parameter Sensitive Analysis

To improve the computational efficiency, we implemented each function $f_i(\cdot)$ for $i = 1, ..., d$ and $g_{k,i}(\cdot)$ for $i = 1, ..., d$ and $k = 1, ..., N$ as an independent MLP. Each MLP takes a $d-$dimensional input, processes it through a hidden layer with $H$ units, and produces a 1-dimensional scalar output. The output of $f_i(\cdot)$ and $g_{k,i}(\cdot)$ are then aggregated by $h_{k,i}(\cdot)$, which we implemented by summation in our reported experiments. The entire architecture consists of $d$ $f_i(\cdot)$ components and $N \times d$ $g_{k,i}(\cdot)$ components, thus totaling $d \times (1 + N)$ distinct MLPs. Given this structure, the per-step complexity is $O(HNd^2)$. For a sequence of length $T$, this results in a total time complexity of $O(THNd^2)$, while the space complexity remains $O(HNd^2)$, as the number of parameters is independent of $T$. We also conducted an empirical experiment to evaluate computational efficiency, setting $d = 100$ and $T = 1000$. The results are presented in the following table. In theory, the "non-degenerate range" refers to regularization strengths that are neither excessively large (which may incorrectly shrink true parent edges) nor excessively small (which may fail to remove non-parent edges). In practice, we find that the method is not sensitive to these hyperparameters as long as they are chosen within a broad and reasonable range. Specifically, across all experiments we set the latent-confounder penalty $\lambda_z$ in the range $[0.1, 0.5]$ (we use values $\{0.1, 0.2, 0.3, 0.35, 0.5\}$) and the intervention-related penalty $\alpha$ in the range $[10^{-3}, 10^{-2}]$, and obtain stable performance throughout these intervals without the need for fine-tuning.

### A.4 PROXIMAL GRADIENT DESCENT

We rewrite the objective in Equation (13) as a sum of a smooth and a non-smooth structured penalty. We first define

$$L(W, Z) = \sum_{k=1}^{N} \sum_{i=1}^{d} \sum_{t=1}^{T} \left\| X_{k,t+1}^i - h_{k,i}\big(f_i(P_{k,t}; W_0^i), g_{k,i}(X_{k,t}; W_k^i)\big) \right\|_2^2, \tag{15}$$

and the regularizer

$$\Omega(W, Z) = \lambda_z \sum_{k=1}^{N} \sum_{l=1}^{L} \sqrt{\frac{1}{T} \sum_{t=1}^{T} Z_{k,l,t}^2}$$

$$+ (1 - \alpha) \sum_{i=1}^{d} \sum_{j=1}^{d+p} \left\| (W_{0,j}^i, W_{1,j}^i, \ldots, W_{N,j}^i) \right\|_2 + \alpha \sum_{k=1}^{N} \sum_{i=1}^{d} \sum_{j=1}^{d} \left\| W_{k,j}^i \right\|_2. \tag{16}$$

Thus the optimization problem can be written as

$$\min_{W, Z} \; L(W, Z) + \Omega(W, Z). \tag{17}$$

At iteration $r$, proximal gradient descent performs one gradient step on the smooth term $L(W, Z)$ followed by a proximal mapping with respect to $\Omega(W, Z)$. We use a common stepsize $\eta > 0$ for simplicity. For each $(k, l)$, we collect the temporal coefficients as a vector

$$Z_{k,l,:}^{(r)} = \big( Z_{k,l,1}^{(r)}, \ldots, Z_{k,l,T}^{(r)} \big) \in \mathbb{R}^T. \tag{18}$$

The corresponding gradient step is

$$\tilde{Z}_{k,l,:}^{(r)} = Z_{k,l,:}^{(r)} - \eta \, \nabla_{Z_{k,l,:}} L\big( W^{(r)}, Z^{(r)} \big), \tag{19}$$

and the proximal update with respect to the group norm $\lambda_z \sqrt{\frac{1}{T} \sum_{t=1}^{T} Z_{k,l,t}^2} = \frac{\lambda_z}{\sqrt{T}} \| Z_{k,l,:} \|_2$ is given by the group soft-thresholding operator

$$Z_{k,l,:}^{(r+1)} = \mathcal{S}_{\tau_z}\big( \tilde{Z}_{k,l,:}^{(r)} \big), \qquad \tau_z = \eta \frac{\lambda_z}{\sqrt{T}}, \tag{20}$$

where for any $v \in \mathbb{R}^m$ and $\tau > 0$,

$$\mathcal{S}_\tau(v) = \begin{cases} \left( 1 - \dfrac{\tau}{\|v\|_2} \right) v, & \|v\|_2 > \tau, \\ 0, & \|v\|_2 \le \tau. \end{cases} \tag{21}$$

For each pair $(i, j)$, we consider the block

$$\mathbf{w}_j^i := \big( W_{0,j}^i, W_{1,j}^i, \ldots, W_{N,j}^i \big), \tag{22}$$

which appears in the structured regularizer

$$(1 - \alpha) \left\| \mathbf{w}_j^i \right\|_2 + \alpha \sum_{k=1}^{N} \left\| W_{k,j}^i \right\|_2. \tag{23}$$

Let the gradient step be

$$u_{k,j}^{i,(r)} = W_{k,j}^{i,(r)} - \eta \, \nabla_{W_{k,j}^i} L\big( W^{(r)}, Z^{(r)} \big), \qquad k = 0, 1, \ldots, N, \tag{24}$$

and denote $\mathbf{u}_j^{i,(r)} = (u_{0,j}^{i,(r)}, \ldots, u_{N,j}^{i,(r)})$. Following the derivation of the proximal mapping for the nested group penalty, we first apply the inner group-wise shrinkage

$$S_{\alpha\lambda\eta}(\mathbf{u}_j^{i,(r)}) = \begin{bmatrix} u_{0,j}^{i,(r)} \\ u_{1,j}^{i,(r)} - \alpha\lambda\eta \dfrac{u_{1,j}^{i,(r)}}{\|u_{1,j}^{i,(r)}\|_2} \\ \vdots \\ u_{N,j}^{i,(r)} - \alpha\lambda\eta \dfrac{u_{N,j}^{i,(r)}}{\|u_{N,j}^{i,(r)}\|_2} \end{bmatrix}, \tag{25}$$

and then apply an outer block soft-thresholding on the whole vector. If $\|S_{\alpha\lambda\eta}(\mathbf{u}_j^{i,(r)})\|_2 \leq (1 - \alpha)\lambda\eta$, the entire block is set to zero. Otherwise, the proximal update is

$$\mathbf{w}_j^{i,(r+1)} = \left(1 - \frac{(1-\alpha)\lambda\eta}{\|S_{\alpha\lambda\eta}(\mathbf{u}_j^{i,(r)})\|_2}\right) S_{\alpha\lambda\eta}(\mathbf{u}_j^{i,(r)}). \tag{26}$$

Equivalently, we can summarize the proximal mapping for each block as

$$\mathbf{w}_j^{i,(r+1)} = \left(1 - \frac{(1-\alpha)\lambda\eta}{\max\left(\|S_{\alpha\lambda\eta}(\mathbf{u}_j^{i,(r)})\|_2, (1-\alpha)\lambda\eta\right)}\right) S_{\alpha\lambda\eta}(\mathbf{u}_j^{i,(r)}), \tag{27}$$

which simultaneously enforces sparsity at both the shared and environment-specific levels.

In the setting of Eq. (13), proximal gradient descent (PGD) has two key advantages over Adam. First, PGD explicitly minimizes the composite objective $L(W, Z) + \Omega(W, Z)$ and, through the proximal mappings in Eq.(20) and Eq.(26), yields exact group-sparse solutions in both the temporal gates $Z$ and the shared/environment-specific weights $W$. This allows us to reliably identify which time-varying latent factors and which covariates are active across environments, leading to interpretable and structurally sparse models. In contrast, Adam (even with standard weight decay) does not correspond to optimizing $L + \Omega$ and typically produces dense parameters, so that temporal and cross-environment structures must be inferred only from small but nonzero coefficients or from ad-hoc pruning, which is not equivalent to solving the regularized problem. On the other hand, Adam often converges faster and may attain slightly better predictive accuracy when structural sparsity is not required. Therefore, PGD is preferable when the primary goal is to learn interpretable, group-sparse $(W, Z)$, while Adam is better suited for purely predictive training without explicit structural constraints.

