# OpenReview forum: "InvarGC: Invariant Granger Causality for Heterogeneous Interventional Time Series under Latent Confounding"
_ICLR.cc/2026/Conference — Submitted to ICLR 2026_

### Official Review · Reviewer_B3Nt · 2025-10-31

**Soundness:** 2
**Presentation:** 2
**Contribution:** 2
**Rating:** 4
**Confidence:** 3

**Summary:**

This paper develops a model that generalizes nonlinear granger causality to a setup where there are unobserved latent confounders and unknown interventions. The current methods in the field of nonlinear granger causality relies on the assumption of causal sufficiency, and the current methods of causal discovery including latent confounders do not handle unknown interventions.
The paper introduces the invarGC algorithm along with identifiability theorems, and then test it on synthetic and real world datasets.

**Strengths:**

* As far as I know this paper is the first to extend Granger Causality to the case where there are nonlinear relations, unobserved latent confounders, and unknown interventions

* This paper gives an algorithm along with an identifiability theorem, an experiment on a toy dataset, and an experiment on an almost real world dataset

* At least equivalent to the tested models for the given task, and better when there are hidden confounders and interventions

**Weaknesses:**

* Some assumptions seem to be stronger than needed and others weaker than needed.

* Many relevant state of the art methods are excluded from the experimental setup

* In the real world dataset for benchmarking, the confounding factors are artificially introduced. The confounding factors are therefore not natural ones.

**Questions:**

* Several state-of-the-art methods which are highly relevant in this setting have not been compared to, such as CD-NOD (Huang et al, 20), RegimePCMCI (Saggioro et al, 2020), JPCMCI (Günther et al, 2023) (these methods assume causal sufficiency similarly to GC, Dynotears, ... but unlike them  they can handle multiple regimes). There is also FCI-JCI (Mooij) which can handle multiple regimes as well as hidden confounding. It is true that FCI-JCI was not introduced  directly for time series but can be easily adapted (using the same strategy as varFCI (Malinsky)) to time series while taking into account instantaneous relations. Is there a reason for not including them?

* Assumptions (1-4) are not explicitly introduced as Assumptions but rather as conditions A1,...,A4 within a Theorem, which is confusing. Later in the text they were refer to as Assumption 1, ..., Assumption 4.

* Don’t you also need the faithfulness assumption to rule out deterministic relationships, not just to exclude path cancellations? (In other words, shouldn't you use the formal definition of Faithfulness which exclude both)

* Since you assume no instantaneous relations and all edges are oriented using time, can't you replace faithfulness with adjacency faithfulness?

Minor:
I’ve always thought of Granger causality as a weaker notion of causality, more about prediction than true causal influence. However, under certain assumptions, doesn’t Granger causality actually correspond to genuine causality? If that’s the case, why do we still use the term Granger causality rather than simply referring to it as causality, given that it is traditionally considered useful mainly for predictive purposes ?

References:

Biwei Huang, Kun Zhang, Jiji Zhang, Joseph Ramsey, Ruben Sanchez-Romero, Clark Glymour, Bernhard Scholkopf.
Causal Discovery from Heterogeneous/Nonstationary Data. JMLR, 2020

Elena Saggioro, Jana de Wiljes, Marlene Kretschmer, Jakob Runge
Reconstructing regime-dependent causal relationships from observational time series. Chaos, 2020


Wiebke Günther, Urmi Ninad, Jakob Runge. Causal Discovery for time series from multiple datasets with latent contexts
UAI, 2023.

Joris M. Mooij, Sara Magliacane, Tom Claassen.
Joint Causal Inference from Multiple Contexts. JMLR, 2020.

Daniel Malinsky, Peter Spirtes.
Causal Structure Learning from Multivariate Time Series in
Settings with Unmeasured Confounding. KDD workshop on causal discovery, 2018.

---

> ### Author Response · Authors · 2025-11-27
> **Author Response to Reviewer B3Nt**
>
> Dear reviewer B3Nt, we appreciate your thoughtful comments and suggestions.
>
> In response to Question 1, we compare our method with PCMCI, which assumes stationarity, no contemporaneous causal links, and no hidden variables, and with LPCMCI, which relaxes this last assumption by allowing hidden variables. We therefore do not include RegimePCMCI or JPCMCI in our experiments; instead, we use CD-NOD as an additional baseline because it is more suitable for our setting, although we observe that its performance degrades when the number of nodes becomes large.
>
> For Question 2, Question 3 and Question 4, thank you for pointing these out. In the standard DAG literature, the usual faithfulness assumption is indeed defined so as to rule out both (i) parameter cancellations along multiple paths and (ii) exact deterministic relationships between variables. In our VAR(1) setting, I agree that what we actually need is a weaker adjacency faithfulness condition: whenever there is a lagged edge $X_t^j \to X_{t+1}^i$, the corresponding causal coefficient is not set to zero in the data-generating process, so that the edge induces a genuine dependence in the population distribution. Because we assume no instantaneous edges and all directions are fixed by time, we do not rely on more delicate conditional independence patterns, and adjacency faithfulness is sufficient in our setting. We have revised Assumption (A4) to make this distinction explicit and avoid confusion with the stronger full Faithfulness assumption, and have updated Assumptions 1–3 accordingly in the revised version.
>
> Discussion on Minor:
> Granger causality is originally defined via predictability and is therefore often viewed as a weaker, prediction-oriented notion, it aligns with true causal influence when certain structural conditions hold. Here we retain the term Granger causality because it refers to the VAR-based, time-lagged causal structure that is naturally suited to multivariate time series. Under standard assumptions such as finite-order dynamics and stationarity, the latter meaning that the same conditional distributions and causal edges repeat over time, identifying the one-step mechanism $X_t \to X_{t+1}$ is sufficient to recover the entire time-unrolled DAG, whose repeated structure is compactly summarized by the Granger causal graph. Although the summary graph itself need not be acyclic, it represents a well-defined causal DAG in the time-expanded domain. Once latent confounding and unknown interventions are properly modeled, the recovered Granger structure again corresponds to the underlying temporal causal mechanism. Conceptually, our contribution is to show that even in the presence of latent confounders and unknown interventions, where classical Granger analysis may no longer be causal, one can recover a Granger causal graph that again admits a causal interpretation once these complications are properly modeled.
>
> We sincerely appreciate your valuable feedback. Thank you again.

---

### Official Review · Reviewer_2Sop · 2025-10-31

**Soundness:** 2
**Presentation:** 2
**Contribution:** 3
**Rating:** 2
**Confidence:** 5

**Summary:**

The paper proposes an algorithm to recover an invariant Granger causal graph from heterogeneous interventional time-series in the presence of latent confounding and unknown intervention targets. Under specific assumptions, the authors establish identifiability of the Granger causal graph, the subspaces spanned by latent confounders, and edge-level interventions.

Methodologically, the approach combines networks for latent-confounder modeling, intervention identification, and invariant predictor learning, and recovers the graph via next-step prediction augmented with some regularizations to enforce invariance and identifiability.

Empirically, on both synthetic and real-world datasets, the method has performance that is competitive with or superior to strong baselines.

**Strengths:**

1. The paper tackles an important gap in the literature by proposing an algorithm that recovers an invariant Granger causal graph under latent confounders and unknown interventions.

2. The paper is well written and logically organized.

3. The work provides both theoretical guarantees and empirical validation.

4. The experimental results show impressive performance, often matching or outperforming the baselines.

**Weaknesses:**

1. The theoretical guarantees are not solid.

1.1 Assumption A4 is vague. Combining the main paper and the appendix, “...interventions are sufficiently diverse to distinguish true causal parents from non-parents.” means that, for $X^j_t\in PA(X^i_{t+1})$, that "there exists at least one environment in which the mechanism of the edge $ j \rightarrow i$ differs from its invariant form.", it also means that for any latent variables connected with the target variable, "there exists an environment in which the mechanism along that direction differs from the invariant one. ".That is, by saying “...interventions are sufficiently diverse to distinguish true causal parents from non-parents,” although the intervention targets are unknown, they must intervene on enough edges so that every parent and latent variable for the target variable is identifiable. This is a very strict assumption.

1.2 With Assumption A3, the latent-to-observed mechanism is invariant. Variables $X^j_t\notin PA(X^i_{t+1})$ induced by such invariant spurious edges could be identified as parents, as there is no edge between $X^j_t$ and $X^i_{t+1}$ and hence such an edge cannot be intervened on, resulting in an invariant spurious edge between $X^j_t$ and $X^i_{t+1}$. I wonder, “by Assumption 4 there exists an environment in which the mechanism along that direction differs from the invariant one,” how this is true for latent variables, as interventions are only for observed variables.

2. Important details about the algorithm are missing.

With finite samples, the hyperparameter $\lambda$, $\alpha$ in equation 12 are important. As claimed in the paper, "For regularization parameters in a standard non-degenerate range, the increase ∆ dominates any penalty saving", what is the practical choice of these hyperparameters? Will the performance be sensitive to these parameters?

3. Scalability and running time are not reported.

It would also be beneficial to clearly list the number of nodes and confounders for each experiment. For instance, what is the number of nodes used for different types of the Causal-Rivers dataset?

4. Ablation results need quantitative clarity.

It would be clearer to include quantitative results demonstrating the decrease in, for instance, AUROC and AUPRC without LCMCI or under other misspecifications. The current visualization is not straightforward, and it is not clear whether it reports a single trial or an average performance. For example, in Figures 3 and 4, what is the ground truth, what is the value in each cell, and why are some cells dark blue even though they are not extremely large or small compared with other cells?

**Questions:**

1. Could you please clarify the meaning of Assumption A4 with a toy example? More generally, what does this assumption require in terms of the number of interventions and the corresponding intervention targets?
2. Are the other baselines also applicable to edge-level interventions? If the setting does not match their assumptions, how does that affect their performance?
3. Is the connection restricted to $Z_t \rightarrow X_{t+1}$ with lag 1? By default, does the confounder connect two variables both at time  $t+1$?
4. Should there be a noise term in equation 1?

---

> ### Author Response · Authors · 2025-11-27
> **Author Response to Reviewer 2Sop-1**
>
> We truly appreciate your careful reading of our paper and the valuable comments. We realized that the main concerns stem from Assumptions 3 and 4, which may have been confusing, and we are grateful for the opportunity to clarify and further improve this part of the paper.
>
> Q1. Could you please clarify the meaning of Assumption...
>
> Here, we would like to clarify that in Assumption 4 (A4), “sufficiently diverse” has two components: (i) sufficiently diverse interventions on the observed variables, and (ii) sufficiently diverse distributional shifts in the latent confounders across environments. First, for each target variable $X_{t+1}^{i}$ and each environment $k$, the environment-specific deviation vector $W_{k}^{i} \in\ R^{d}$ encodes which incoming edges $X_{t}^{j} \to X_{t+1}^{i}$ are modified by interventions. From $W_{k}^{i}$ for $k \in [1,\dots,N]$, we can derive an intervention signature matrix for target $i$: its $(k,j)$-entry is $1$ if the edge $X_t^j \to X_{t+1}^i$ is intervened in environment $k$, and $0$ otherwise. The column corresponding to edge $j$ therefore records in which environments this edge is intervened. If the entire column is zero for a true parent edge, that edge is never intervened in any environment, making it impossible to separate its invariant causal effect from spurious associations at the population level. We say that interventions on the observed variables are sufficiently diverse if, for each target $X_{t+1}^i$: (a) every true parent edge $X_t^j \to X_{t+1}^i$ is intervened in at least one environment (its column is not identically zero), and (b) the intervention signature matrix restricted to the true parent columns has full column rank, so that the intervention patterns of different parents are not redundant. Intuitively, (b) rules out degenerate cases where parents are always intervened in exactly the same way and hence cannot be disentangled.
>
> Second, let $k \in [1,\dots,N]$ index environments and let $P_{k}(Z_t)$ denote the marginal distributions of the latent confounders $Z_t$ in environment $k$. We assume that the latent-to-observed mechanism is invariant across environments
> (as in Assumption 3), i.e.,
> $X_{t+1}= f_X\big(W_{X_{t+1}X_t}^\top X_t + W_{X_{t+1}Z_t}^\top Z_t\big) + \epsilon_{t+1}^X,$
> with the same $f_X$, $W_{X_{t+1}X_t}$, and $W_{X_{t+1}Z_t}$ for all $k$,
> while the marginal distribution of $Z_t$ is allowed to vary with $k$. Formally, write $\mu_k := E_k[Z_t]$ and $\sigma_k := Cov_k(Z_t)$ for the mean and covariance of $Z_t$ in environment $k$. We say that the latent distributions are sufficiently diverse if the collection $P_k(Z_t)$ for $k=1 \dots N$ induces linearly independent variations in the effective confounding term $W_{X_{t+1}Z_t}^\top Z_t$ across environments. A simple sufficient condition is that the set of projected means $W_{X_{t+1}Z_t}^\top \mu_k$ for $k=1 \cdots N$ span the same subspace as $W_{X_{t+1}Z_t}^\top Z_t^\star$, i.e.,
> $span(W_{X_{t+1}Z_t}^\top\mu_1,\dots, W_{X_{t+1}Z_t}^\top \mu_N\) =span(W_{X_{t+1}Z_t}^\top Z_t^\star\).$
> Intuitively, different environments must induce non-redundant shifts in
> the confounders along directions that actually affect $X_{t+1}$.
> Otherwise, environment-induced variation in $Z_t$ could be absorbed into
> the noise and the latent subspace would not be identifiable from $X$ alone.
>
> Q2. With finite samples, the hyperparameter...
>
> In theory, the “non-degenerate range” refers to regularization strengths that are neither excessively large (which may incorrectly shrink true parent edges) nor excessively small (which may fail to remove non-parent edges). In practice, we find that the method is not sensitive to these hyperparameters as long as they are chosen within a broad and reasonable range. Specifically, across all experiments we set the latent-confounder penalty $\lambda_z$ in the range $[0.1, 0.5]$ (we use values $\{0.1, 0.2, 0.3, 0.35, 0.5\}$) and the intervention-related penalty $\alpha$ in the range $[10^{-3}, 10^{-2}]$, and obtain stable performance throughout these intervals without the need for fine-tuning.
>
> Q3. Scalability and running time are not reported.
>
> Thank you for the suggestion. We have now included a comprehensive time complexity analysis in the appendix.
>
> Q4. Ablation results need quantitative clarity.
>
> In Figures 3 and 4, we visualize the learned structures using the corresponding ground truth provided in the appendix. Each cell value is computed from the regularized weights in the first layer, and we apply a threshold based on the order of magnitude. In Figure 4, the three panels from left to right correspond to the recovered latent confounder subspace, the Granger causal structure, and the interventions, respectively, illustrating that the proposed method can jointly recover all three components.

---

> ### Author Response · Authors · 2025-11-27
> **Author Response to Reviewer 2Sop-2**
>
> Q5. Are the other baselines also applicable to edge-level interventions...
>
> Most existing intervention-based methods rely on node-level interventions and further assume causal sufficiency. These assumptions limit their applicability in scenarios where latent confounders are present, as the model cannot distinguish true causal effects from spurious associations induced by unobserved variables.
>
> Q6. Is the connection restricted to...
>
> Yes. In our theoretical model and synthetic data we work with a VAR(1) structure, so all latent-to-observed edges are of the form $Z_t \to X_{t+1}$. The latent variable acts as a common cause for multiple components of $X_{t+1}$ via edges $Z_t \to X_{t+1}^i$ and $Z_t \to X_{t+1}^j$, which induces confounding between $X_{t+1}^i$ and $X_{t+1}^j$. We do not include instantaneous effects such as $Z_{t+1} \to X_{t+1}$ or $Z_t \to X_t$ in the default setting. Extending to higher lags is conceptually straightforward: we can include additional lagged variables in the state vector and apply the same identifiability arguments to this augmented process.
>
> Q7. Should there be a noise term in equation 1?
>
> Yes, thank you for catching this. We have added the missing noise term in Equation (1) in the revised version.

---

### Official Review · Reviewer_xZVy · 2025-11-01

**Soundness:** 3
**Presentation:** 2
**Contribution:** 3
**Rating:** 4
**Confidence:** 3

**Summary:**

The paper considers the problem of Granger causal discovery from time series data. Specifically, the authors consider the presence of both latent confounding and multi-domain heterogeneity, where the causal relations among variables (i.e., the causal graph) are the same across all environments, but the causal mechanism (functional relations) among observed variables may vary. The authors show that, when the model is linear and the lag size is one (i.e., $X_{t+1}$ does not depend on $X_{1:t-1}$ and $Z_{1:t-1}$ conditioned on $X_t$ and $Z_t$), the true causal graph can be uniquely identified, which can be recovered by minimizing the loss function in the proposed recovery algorithm. The authors evaluate the performance of the algorithm on both synthetic and real datasets.

**Strengths:**

1. The problem is well-formulated and considers a very realistic and under-studied setting.
2. The authors conduct extensive simulations to demonstrate the effectiveness of the proposed algorithm, especially on real-world datasets.

**Weaknesses:**

1. The notation is a little bit confusing and hard to follow. For example, the subscript of $W$ includes both numbers and variables. It would be better if it can be unified (say use $W_{0, 1:d}$ instead of $X_{0,X_{t+1}X_t}$).
2. Some of the technical details are not clearly explained, such as mathematical formulation of certain assumptions and technical details (see Q1 and Q4 below).
3. It seems to me that the identification results considers a much simpler setting than the model described in Section 3.2 (see Q5 and Q6 below).

**Questions:**

1. Is there a mathematical formulation of "sufficiently diverse" in (A4)?
2. Are there any restrictions on the "minimality" of interventions in Theorem 3? For example, suppose there are three environments and two observed variables, where environment 1 is invariant and the node $x_2$ is intervened in environments 2 and 3. Then there exists another model where environment 2 is invariant and the node $x_2$ is intervened in environment 1 and 3.
3. In line 359, does "two environments remain purely observational" imply that these two environments share exactly the same causal mechanism?
4. In Equation (7), is LCIM a neural network? If yes, how are the parameters optimized?
5. How do the causal effects among latent variables (i.e., W_{Z_{t+1}Z_t}) affect the performance of the recovery algorithm?
6. Do the theoretical results presented in Section 4.3 only hold in linear setting?

---

> ### Author Response · Authors · 2025-11-27
> **Author Response to Reviewer xZVy-1**
>
> Dear reviewer xZVy, thank you for the insightful comments. We have revised the notations in the paper based on your suggestions and are pleased to address the questions you have raised.
>
> Q1: Is there a mathematical formulation of "sufficiently diverse" in assumption (A4)?
>
> Here, we would like to clarify that in Assumption 4 (A4), “sufficiently diverse” has two components: (i) sufficiently diverse interventions on the observed variables, and (ii) sufficiently diverse distributional shifts in the latent confounders across environments. First, for each target variable $X_{t+1}^{i}$ and each environment $k$, the environment-specific deviation vector $W_{k}^{i} \in\ R^{d}$ encodes which incoming edges $X_{t}^{j} \to X_{t+1}^{i}$ are modified by interventions. From $W_{k}^{i}$ for $k \in [1,\dots,N]$, we can derive an intervention signature matrix for target $i$: its $(k,j)$-entry is $1$ if the edge $X_t^j \to X_{t+1}^i$ is intervened in environment $k$, and $0$ otherwise. The column corresponding to edge $j$ therefore records in which environments this edge is intervened. If the entire column is zero for a true parent edge, that edge is never intervened in any environment, making it impossible to separate its invariant causal effect from spurious associations at the population level. We say that interventions on the observed variables are sufficiently diverse if, for each target $X_{t+1}^i$: (a) every true parent edge $X_t^j \to X_{t+1}^i$ is intervened in at least one environment (its column is not identically zero), and (b) the intervention signature matrix restricted to the true parent columns has full column rank, so that the intervention patterns of different parents are not redundant. Intuitively, (b) rules out degenerate cases where parents are always intervened in exactly the same way and hence cannot be disentangled.
>
> Second, let $k \in [1,\dots,N]$ index environments and let $P_{k}(Z_t)$ denote the marginal distributions of the latent confounders $Z_t$ in environment $k$. We assume that the latent-to-observed mechanism is invariant across environments
> (as in Assumption 3), i.e.,
> $X_{t+1}= f_X\big(W_{X_{t+1}X_t}^\top X_t + W_{X_{t+1}Z_t}^\top Z_t\big) + \epsilon_{t+1}^X,$
> with the same $f_X$, $W_{X_{t+1}X_t}$, and $W_{X_{t+1}Z_t}$ for all $k$,
> while the marginal distribution of $Z_t$ is allowed to vary with $k$. Formally, write $\mu_k := E_k[Z_t]$ and $\sigma_k := Cov_k(Z_t)$ for the mean and covariance of $Z_t$ in environment $k$. We say that the latent distributions are sufficiently diverse if the collection $P_k(Z_t)$ for $k=1 \dots N$ induces linearly independent variations in the effective confounding term $W_{X_{t+1}Z_t}^\top Z_t$ across environments. A simple sufficient condition is that the set of projected means $W_{X_{t+1}Z_t}^\top \mu_k$ for $k=1 \cdots N$ span the same subspace as $W_{X_{t+1}Z_t}^\top Z_t^\star$, i.e.,
> $span(W_{X_{t+1}Z_t}^\top\mu_1,\dots, W_{X_{t+1}Z_t}^\top \mu_N\) =span(W_{X_{t+1}Z_t}^\top Z_t^\star\).$
> Intuitively, different environments must induce non-redundant shifts in
> the confounders along directions that actually affect $X_{t+1}$.
> Otherwise, environment-induced variation in $Z_t$ could be absorbed into
> the noise and the latent subspace would not be identifiable from $X$ alone.
>
> Q2: Are there any restrictions on the "minimality" of interventions in Theorem 3?
>
> Yes. In Theorem 3 we implicitly assume that interventions are sparse, i.e., the invariant mechanism is the majority pattern and interventions occur only in a minority of environments. Concretely, for each edge $X_t^j \to X_{t+1}^i$, let $W_{k,j}^i$ denote its deviation parameter in environment $k$. We assume that the index set $S_{i,j} := [k \in [1,\dots,N] : W_{k,j}^i \neq 0] $ satisfies$ |S_{i,j}| < N/2$. We have added this clarification in the revised version of the paper.
>
> Q3: In line 359, does "two environments remain purely observational"...?
>
> Yes. It means that the two environments are generated from the same underlying structural model with identical causal mechanisms (i.e., the same Granger coefficients and latent-to-observed parameters); they differ only through independent noise realizations (and random initial states). The third environment is then obtained by applying edge-level interventions that modify a subset of these coefficients.

---

> ### Author Response · Authors · 2025-11-27
> **Author Response to Reviewer xZVy-2**
>
> Q4: In Equation (7), is LCIM implemented as a neural network? If so, how are the parameters optimized?
>
> In our implementation, for each environment and time step, we introduce a learnable latent vector $Z_{k,t}$ and concatenate it with the observed variables $X_{k,t}$; this concatenated $[X_{k,t}, Z_{k,t}]$ is fed directly into the invariant Granger causal network. The latent vectors and all network weights are optimized jointly. We additionally include an explicit regularization term on $Z$ in Eq.(13), which encourages the model to use the latent variables only when necessary and thereby helps recover a meaningful latent confounder subspace together with the underlying causal relations.
>
> Q5: How do the causal effects among latent variables (i.e., $W_{Z_{t+1}Z_t}$) affect the performance of the recovery algorithm?
>
> In our theory, as long as the latent process is stable and exogenous as in Assumption (A3), $W_{Z_{t+1}Z_t}$ does not affect which quantities are identifiable. In our experiments, if its entries are very small, the latent process is close to i.i.d.noise and the confounding signal is weak, so more data are needed to reliably recover the latent subspace. If its entries are very large, the latent process becomes very persistent, which can make optimization harder and also require longer time series. We therefore use moderate values in the synthetic experiments, so that latent confounding is clearly present while the latent dynamics remain stable.
>
> Q6: Do the theoretical results presented in Section 4.3 only hold in the linear setting?
>
> Yes. Our formal identifiability results are proved for the linear structural time series model. The nonlinear formulation in Eq. (13) and our neural implementation go beyond this linear model theory and are supported empirically; extending the proofs to general nonlinear function classes is left as future work.

---

### Official Review · Reviewer_64gK · 2025-11-01

**Soundness:** 3
**Presentation:** 2
**Contribution:** 2
**Rating:** 4
**Confidence:** 3

**Summary:**

This paper proposes InvarGC, a framework for Granger causality discovery in heterogeneous interventional time series subject to latent confounding. InvarGC identifies invariant causal relations by using data heterogeneity across environments, infers latent confounders through a dedicated inference module, and distinguishes intervened from non-intervened environments at the edge level. The authors offer formal identifiability guarantees of the recovered causal graph. The authors also conduct comprehensive experiments with synthetic and real-world datasets to support their claims.

**Strengths:**

- Tackles a relevant setting: unknown interventions + latent confounding, and the problem setup is clear.
- Identifiability results (graph, latent subspace, edge-level interventions) with explicit assumptions.
- Good experimental results vs. strong baselines across synthetic and real data; sensible ablations on $L$.

**Weaknesses:**

1. The largest real-world example (TEP) uses 33 variables, and Causal-Rivers uses node subsets. While nontrivial, this leaves open whether InvarGC scales effectively to higher-dimensional (>100 variables), longer sequences, or truly networked time series encountered in domains such as neuroscience, genomics, or industrial process control. No runtime or computational complexity results are reported either.
2. Although the ablation study in Figure 3 analyzes the effect of the number of latent confounders ($L$) and regularization weights, the empirical analysis is somewhat superficial. There is insufficient exploration of how robust the method is to hyperparameter misspecification in practice, especially under lackluster prior knowledge of the true confounder count. It is unclear how challenging tuning becomes as the dataset grows, or whether the method is stable across a realistic hyperparameter grid.
3. Edge-level detection uses an ad-hoc threshold; no uncertainty or sensitivity analysis.

**Questions:**

1. What are training/runtime and memory costs as $d$ and $T$ grow?
2. Can you provide more details on the choice and parameterization of the non-linear functions $f_i(\cdot)$, $g_{k,i}(\cdot)$, and $h_{k,i}(\cdot)$? Are these always neural networks, and how sensitive are your results to their depth/width or activation choices?
3. Is edge-level intervention detection always threshold-based? Would a probabilistic approach or inclusion of uncertainty quantification improve detection stability, particularly for weak interventions?

---

> ### Author Response · Authors · 2025-11-27
> **Author Response to Reviewer 64gK**
>
> Dear reviewer 64gK, thank you so much for your suggestions and comments. We are glad to have the opportunity to provide clarifications and answers to your questions.
> 1. For questions 1 and 2 "What are training/runtime..." and "More details on the non-linear functions...":
>
> To improve the computational efficiency, we implemented each function $f_i(\cdot)$ for $i=1,...,d$ and $g_{k,i}(\cdot)$ for $i=1,...,d$ and $k=1,...,N$ as an independent MLP. Each MLP takes a $d-$dimensional input, processes it through a hidden layer with $H$ units, and produces a 1-dimensional scalar output. The output of $f_i(\cdot)$ and $g_{k,i}(\cdot)$ are then aggregated by $h_{k,i}(\cdot)$, which we implemented by summation in our reported experiments. The entire architecture consists of $d$ $f_i(\cdot)$ components and $N \times d$ $g_{k,i}(\cdot)$ components, thus totaling $d \times (1+N)$ distinct MLPs. Given this structure, the per-step complexity is $O(HNd^2)$. For a sequence of length $T$, this results in a total time complexity of $O(THNd^2)$, while the space complexity remains $O(HNd^2)$, as the number of parameters is independent of $T$.
>
> 2. For question 3 "Is edge-level intervention detection always threshold-based? Would a probabilistic approach...":
>
> Our original model was trained with the Adam optimizer, and the causal structure was inferred from the first-layer parameters. As a result, we previously applied a post-hoc threshold to identify interventions. To resolve this issue, we developed a new algorithm based on proximal gradient descent with group soft-thresholding. This approach naturally enforces sparsity by driving entire parameter groups to exactly zero during training, thereby removing the need for manual thresholding. We have updated the algorithm and discussed the pros and cons of the two implementations in the appendix accordingly.
>
> If you have any further questions, please feel free to let us know. We would be happy to discuss them. Thank you for your time!

---

> > ### Comment · Reviewer_64gK · 2025-11-27
> >
> > I thank the authors for their detailed response and the updates to the manuscript. They have adequately addressed my concerns and so I am increasing my score.

---

### Meta-Review · Area_Chair_R3Gk · 2026-01-06

**Summary:**

All the reviewers unanimously raise concerns about the clarity and strength of the assumptions and whether they are realistic, especially Assumption A4 (“sufficiently diverse” interventions). The reviewers have also raised comments and questions on whether the identifiability results are aligned with the nonlinear model used in practice, whether faithfulness assumptions are correctly specified, and whether latent–latent dynamics could undermine recovery. Some of the reviewers have also raised technical questions on the lack of sufficient clarity on the neural architecture, optimization, thresholding for edge-level intervention detection, hyperparameter sensitivity, and missing noise terms. Furthermore, the reviewers have concerns about the lack of sufficient evidence on scalability and complexity.

The authors' responses to some extent clarity some of the issues raised. For instance, the manuscript now formalizes Assumption A4 in the appendix. While this clarity is welcome, at the same time reassures that the assumption is indeed strong. The paper revisions also correctly narrow the theory to linear VAR(1) with adjacency faithfulness, aligning claims with what is actually proved, but this also underscores that nonlinear guarantees remain empirical. Algorithmic clarifications. On the empirical side, claims of hyperparameter robustness and scalability seem to have scopes limited for the claims made. Overall, the rebuttal strengthens the rigor of the analysis, but it also implies the main criticism of the reviewers, that is the identifiability guarantees reliy on restrictive assumptions and that empirical validation at scale is still limited.

**Reviewer Concerns:**

The comments pertinent to the clarity of the assumptions and exposition are addressed. These, on the other hand, confirm the reviwers' concern that some of the assumptions are unrealistically strong.

**Reviewer Scores:**

In the absence of follow-up discussion from the reviewers, it isn't easy to gauge how they would have adjusted their ratings. Nevertheless, from the discussions, it's expected that the reviewers will not increase their ratings significantly, without which the paper does not rise to the level of acceptability.

---

### Decision · Program_Chairs · 2026-01-26

Reject